# Class-Adaptive Rectification with Experts for Robust Long-Tailed Noisy Label Learning

## Abstract

Real-world datasets frequently exhibit long-tailed class distributions alongside noisy labels, posing compounded challenges for robust learning. While recent methods have made progress, they often neglect the uneven impact of label noise across classes, resulting in insufficient correction for tail classes. This imbalance further introduces erroneous over-regularization on other classes, ultimately undermining long-tailed learning. To address these challenges, we propose Class-Adaptive Rectification with Experts (CARE), a parameter-efficient framework built upon vision–language models, which performs class-aware label correction by jointly leveraging three complementary sources of supervision: noisy observed labels, text embeddings, and image features. CARE further employs a class-adaptive Top-$K$ expert consensus mechanism, which assigns smaller $K$ to tail classes in order to extract reliable candidate labels and recalibrate class frequencies. This refinement yields faithful class-frequency estimation, thereby enabling more reliable long-tailed calibration. We evaluate CARE on CIFAR-100-LTN, mini-ImageNet-LTN, and real-world datasets, including Food101N and WebVision-50. Across all benchmarks, CARE consistently surpasses recent state-of-the-art methods, achieving up to 3.0% accuracy improvements in certain settings. The source code is temporarily available at https://anonymous.4open.science/r/CARE-9F10.

## 1 Introduction

Deep learning models rely heavily on large-scale datasets that are not only accurately annotated but also approximately class-balanced to ensure robust generalization (Menon et al., 2021). The rise of large-scale foundation models in recent years has further amplified this demand for high-quality labeled data (Dong et al., 2023; Li et al., 2024a), as insufficient label quality may introduce unintended prior biases during training. However, such ideal conditions are rarely met in real-world scenarios. Accurate labeling is often prohibitively time-consuming and labor-intensive, while data acquisition pipelines inherently introduce imbalance due to varying collection difficulty across classes (De Angeli et al., 2022; Hoyer et al., 2022). Rare or under-represented categories, often referred to as tail classes, are particularly difficult to collect, leading to extreme data skew (Wang et al., 2017). Consequently, real-world datasets are inherently challenged by the co-occurrence of pervasive annotation errors (noisy labels (Xiao et al., 2015)) and extreme class imbalance (long-tailed distributions (Zhang et al., 2023b)), which synergistically degrade model generalization and severely impair recognition performance, especially on tail classes. When tackled separately, long-tail learning methods assume clean labels and thus may amplify mislabeling in scarce tail classes, while noisy-label learning methods assume roughly balanced classes and tend to discard noisy samples, thereby losing valuable information in under-represented categories (Zhang et al., 2024).

To address the practical and pervasive challenge of learning from data with both label noise and long-tailed distributions, the concept of long-tailed noisy label (LTNL) learning has recently been introduced (Lu et al., 2023; Zhang et al., 2023a). Several recent studies have explored this joint setting, with some methods incorporating long-tail characteristics into the noise detection process. A common strategy involves first identifying noisy samples, followed by applying long-tailed learning techniques (Yi et al., 2022; Zhang et al., 2023a). However, such sequential pipelines critically depend on the performance of the specifically designed noise detectors, which are typically tailored to either noise robustness or long-tailed data, limiting their effectiveness in addressing both issues simultaneously. For example, Chen et al. (2025) propose a Robust Logit Adjustment (RLA), which

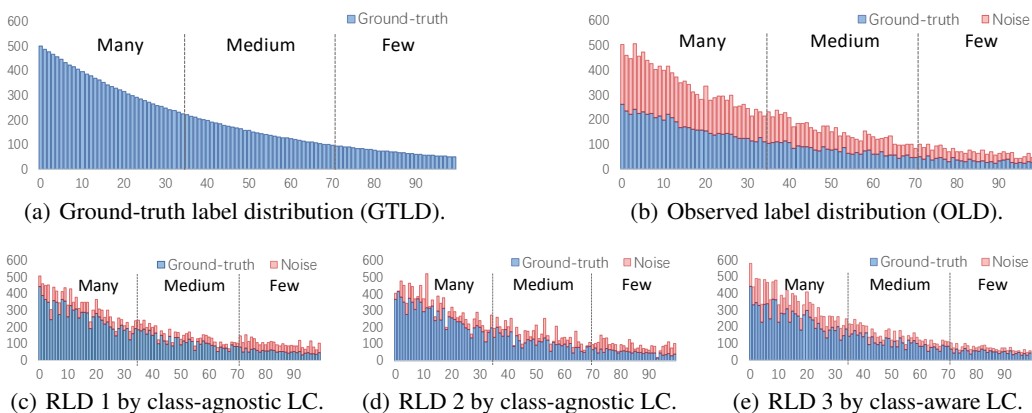

Figure 1: Class-wise label distribution comparison on CIFAR-100. (a) shows the ground-truth distribution with an imbalance factor of 10. (b) presents the distribution under 50% joint noise. (c–e) show rectified label distributions (RLDs) obtained by different label correction (LC) methods.

first corrects the noisy labels and then leverages the refurbished label distribution for logit adjustment, yet still relies on a class-agnostic label correction strategy. Moreover, most existing methods simplify the problem by assuming a uniform noise rate across classes, ignoring the more realistic scenario where tail classes tend to exhibit higher noise levels. Lu et al. (2023) take a more flexible view by considering discrepancies between the observed and intrinsic class distributions, highlighting the need for more flexible and class-aware solutions.

Table 1: Acc. (%) comparison across different rectified labels.

| Lable distribution | $NR$ | Head | Med. | Tail | All |
|---|---|---|---|---|---|
| GTLD (Figure 1(a)) | 0 | 86.2 | 84.9 | 82.9 | 84.8 |
| OLD (Figure 1(b)) | 50% | 79.6 | 79.9 | 78.8 | 79.5 |
| RLD 1 (Figure 1(c)) | 24% | 80.6 | 79.7 | 78.2 | 79.6 |
| RLD 2 (Figure 1(d)) | 25% | 77.0 | 76.5 | 77.0 | 76.8 |
| RLD 3 (Figure 1(e)) | 28% | 81.4 | 80.9 | 80.2 | 80.9 |

Although class-agnostic label refurbishment methods show promise in reducing overall noise, their effectiveness under long-tailed scenarios is limited and can even degrade performance in certain cases. In particular, we preliminarily adopted the label correction strategy from Chen et al. (2025), generating two rectified label distributions under different hyperparameters, as illustrated in Figure 1(c) and Figure 1(d). For comparison, we also present the ground-truth label distribution, the initial 50% joint noise distribution, and the CARE-refined class-aware label distribution in Figure 1(a), Figure 1(b), and Figure 1(e), respectively. Fine-tuning CLIP with these three types of noisy-labeled data yields the results shown in Table 1. Despite achieving similar overall noise rates and significantly reducing label noise compared to the original noisy labels, the results reveal that insufficient correction of tail-class labels (e.g., Figure 1(c)) yields limited performance gains (79.6% vs. 79.5% ), or even leads to degradation, as seen in Figure 1(d) vs. Figure 1(b) (total accuracy 76.8% vs. 79.5%). The decline stems from inaccurate regularization introduced during long-tailed learning. This observation shows that merely lowering the noise ratio is not sufficient, and class-aware rectification is essential. In contrast, by emphasizing accurate tail label correction, RLD 3 (shown in Figure 1(e)) ensures that the strong calibration induced by tail classes is meaningful, while the weaker regularization from uncorrected head labels avoids over-regularization and/or over-calibration. This class-aware strategy leads to more balanced supervision and improves overall robustness.

To this end, we propose a novel framework named Class-Adaptive Rectification with Experts (CARE) for robust learning under long-tailed noisy label (LTNL) settings. CARE leverages three complementary sources of information, namely, observed labels, Vision-Language Model (VLM)-based text embeddings, and VLM-based image features, to derive class-adaptive expert consensus for robust label rectification. Without introducing additional parameters, CARE identifies low-noise candidates through adaptive class-wise Top-$K$ voting, where larger $K$ values are used for head classes and smaller $K$ for tail classes to reflect their respective data abundances and reduce confirmation bias in underrepresented categories. The expert-consensus semantics are then dynamically integrated into class frequency estimation, while low-confidence predictions are filtered through a strict hierarchical filtering mechanism that enforces consistency across classes. This enables CARE to produce reliable label distributions and perform effective noise correction. Our main contributions are summarized as follows: 1) We identify that ignoring tail label noise, while significantly reducing overall noise rate, fails to improve overall performance under long-tailed distributions. 2) We propose

a parameter-free CARE framework that unifies noisy label correction and long-tailed calibration for improved generalization. 3) We design a class-aware consensus mechanism that dynamically adjusts correction confidence based on class frequency and inter-expert agreement. 4) Extensive experiments on both synthetic and real-world LTNL benchmarks demonstrate that CARE consistently outperforms state-of-the-art methods under various noise and imbalance settings.

## 2 PROPOSED METHOD

### 2.1 PRELIMINARIES

**Basic Notations.** Given a $C$-class classification problem, where the training set is denoted as $\mathcal{D}_{train} = \{(x_i, \tilde{y}_i)\}_{i=1}^N$, with $x_i$ representing an input image and $\tilde{y}_i \in \tilde{\mathcal{Y}}$ being the corresponding observed label. Due to annotation noise, the observed label set $\tilde{\mathcal{Y}}$ may differ from the unknown ground-truth label set $y \in \mathcal{Y}$, and such discrepancies are regarded as noisy labels. Without loss of generality, classes are indexed in descending order of sample frequency $n_i$, i.e., $n_1 > n_2 > \ldots > n_C$, with $n_1 \gg n_C$, characterizing the long-tailed nature of the data distribution. Throughout this paper, subscripts are used to indicate sample or class indices.

**Problem Analysis.** In long-tailed noisy-label learning, the combination of sample scarcity and annotation noise severely degrades tail-class performance (Sheng et al., 2024b). Existing methods often apply class-agnostic corrections that fail to distinguish intrinsic uncertainty (from few samples) and actual noise, leading to over-filtering of tail data and biased learning (Shu et al., 2025; Zhang et al., 2023a; Lu et al., 2023). Moreover, long-tail rebalancing techniques may amplify the impact of mislabeled tail samples (Menon et al., 2021; Li et al., 2023). These issues hinder both label refinement and distribution calibration (see Appendix A for detailed analysis). We propose **CARE** that refines noisy labels by computing high-confidence consensus across multiple experts. By explicitly enhancing tail-class label reliability and disentangling class uncertainty from annotation noise, CARE enables more accurate and robust long-tail distribution calibration.

### 2.2 METHOD OVERVIEW.

CARE integrates three complementary sources of information to estimate reliable class distributions: 1) *Text Expert (TE)*. A CLIP-based text encoder (Radford et al., 2021) predicts class confidences from the semantic description of each label (no additional fine-tuning). 2) *Image Expert (IE)*. A CLIP-based image encoder generates visual-semantic class confidences for each input image, fine-tuned via AdaptFormer (Chen et al., 2022) for downstream tasks. 3) *Observed-label expert* is referred to as Base Expert (BE). BE utilizes the original (noisy) annotations, which nonetheless retain valuable class-indicative information. CARE aggregates class-wise confidence scores from all three experts using a relevance-weighted strategy to compute a consensus distribution. This yields a refined label distribution that simultaneously 1) down-weights unlikely noisy assignments in tail classes, 2) preserves high-confidence head-class labels, and 3) approximates the true underlying class frequencies. The resulting corrected labels exhibit substantially reduced noise rates and a distribution closer to the intrinsic (clean) data distribution. Finally, standard long-tail calibration (e.g., logit adjustment (Menon et al., 2021)) can be applied.

### 2.3 COMPOSITION OF PARAMETER-FREE EXPERTS

This approach integrates signals from three distinct experts (text, image, and observed-label), leveraging their complementary strengths to identify high-confidence candidate classes, rather than relying on a single source.

**Text Expert (TE).** We leverage the text encoder of a pre-trained VLM, such as CLIP (Radford et al., 2021), to embed each class label into a shared visual-semantic space and obtain $\{\mathbf{t}_c\}_{c=1}^C$. This encoder captures rich prior knowledge from large-scale pre-training and aligns textual semantics with visual representations. By computing the similarity between embedded label texts and image features, the model can estimate class-wise confidence scores without requiring additional fine-tuning. This allows TE to provide robust semantic guidance and highlight potentially correct classes, unaffected by the corrected observed labels. For an input image $x$[1], TE provides the confidence vector $\mathbf{p}^{TE} = \{p_c^{TE}\}$ based on the cosine similarity between the general image feature $\hat{\mathbf{f}}$ (extracted from the pretrained

---

[1]We omit subscript notation for clarity.

image encoder) and the corresponding text features $\mathbf{t}_c$:

$$p_c^{TE} = \frac{\exp(z_c^{\text{TE}})}{\sum_{j=1}^{C} \exp(z_j^{\text{TE}})}, \text{ with } z_c^{\text{TE}} = s \cdot \frac{\mathbf{t}_c^\top \hat{\mathbf{f}}}{\|\mathbf{t}_c\|\|\hat{\mathbf{f}}\|}, \tag{1}$$

where $s$ is the scale parameter. The softmax-normalized similarity yields a semantic prior over class labels based purely on text-image alignment, without additional fine-tuning.

**Image Expert (IE).** The image expert is derived from the fine-tuned CLIP image encoder, which produces task-adapted visual representations $\mathbf{f}$ for each input image $x$. To enhance adaptability to the downstream tasks while maintaining efficiency, we adopt AdaptFormer (Chen et al., 2022) for parameter-efficient fine-tuning. It is worth noting that this fine-tuning is performed solely for the final classification purposes, and the resulting adapted image encoder is reused for label correction without introducing additional training objectives. IE provides the confidence vector $\mathbf{p}^{\text{IE}} = \{p_c^{\text{IE}}\}$ by the similarity between $\mathbf{f}$ and the class-specific weights $\mathbf{w}_c$ of the linear classifier $\mathbf{W}$:

$$p_c^{\text{IE}} = \frac{\exp(z_c^{\text{IE}})}{\sum_{j=1}^{C} \exp(z_j^{\text{IE}})}, \quad \text{where} \quad z_c^{\text{IE}} = s \cdot \frac{\mathbf{w}_c^\top \mathbf{f}}{\|\mathbf{w}_c\|\|\mathbf{f}\|}. \tag{2}$$

IE leverages the fine-tuned visual representation tailored to the target domain and serves as a byproduct of the main training pipeline. It remains parameter-free during the expert consensus phase.

**Observed-Label Expert (BE)** The observed-label expert, namely the base expert, relies directly on the original (potentially noisy) label annotations. Despite their unreliability, these labels still contain valuable supervision, especially for head classes that contain a large number of accurate labels. BE treats the observed label as an indicator of class assignment and serves as a complementary weak signal in the consensus-building process. The observed label $\hat{y}$ is represented as a one-hot confidence vector $\mathbf{p}^{BE} = \{p_c^{BE}\}$, where only the $\hat{y}$-th entry is set to 1.

## 2.4 CLASS AWARE EXPERT CONSENSUS ACCUMULATION

To address the limitations of class-agnostic label refurbishment, particularly its inability to differentiate between head and tail class noise, we introduce a class-aware expert consensus mechanism. We adopt a consensus strategy that accumulates class-wise support across experts, effectively refining the label distribution with an awareness of class frequency.

**Class Aware Expert Consensus.** To evaluate the reliability of the observed label, we first identify the top-$K$ most confident predictions from both TE and IE:

$$\mathcal{T}_K^* = \text{TopK}(\mathbf{p}^*), \tag{3}$$

where $* \in \{\text{TE}, \text{IE}\}$, $\mathbf{p}^*$ denotes the confidence scores from the respective expert. To improve robustness against noisy labels and long-tailed class distributions, we introduce a class-adaptive Top-$K$ consensus mechanism. The number of top predictions considered, $K$ is adaptively determined based on the frequency of each class. Specifically, classes with higher sample counts (head classes) are assigned larger $K$, while those with fewer samples (tail classes) are assigned smaller $K$, allowing the model to allocate trust more appropriately across the class spectrum. Formally, the class-specific $K$ is defined to be proportional to the class frequency: $K_c \propto n_c^e$, where $n_c^e$ denotes the sample count for class $c$ at epoch $e$. At the initial stage ($e = 0$), $n_c^0$ is computed based on the observed labels. For $e > 0$, $n_c^e$ is updated using the corrected labels from the previous training epoch.

**Class Frequency Accumulation.** For each sample $x$ and class $c$, the accumulated frequency is updated by:

$$F_c^{(e)} = F_c^{(e-1)} + \sum_{m \in \{\text{TE, IE, BE}\}} \alpha_m(x)\, g_m(c), \tag{4}$$

where $m$ indexes the experts (TE, IE, BE). The class-wise contribution of expert $m$ is:

$$g_m(c) = \mathbb{I}[c \in \mathcal{T}_K^m]\, p_c^m, \tag{5}$$

meaning that expert $m$ contributes confidence only for classes in its top-$K$ list. The reliability weight assigned to expert $m$ for sample $x$ is:

$$\alpha_m(x) = \begin{cases} \sum_{j \in \mathcal{T}_K^m} p_j^m, & \text{if } \tilde{y} \in \mathcal{T}_K^m, \\ 1, & \text{otherwise.} \end{cases} \tag{6}$$

Figure 2: Overview of CARE. Different colors in the label denote classes, and intensity reflects confidence. ① illustrates expert consensus (TE and BE agree on Top-1 of TE). The top-1 confidence of TE that is refined by removing unlikely classes is assigned to the consensus class (blue). Top-2 and Top-3 confidences, lacking consensus, are individually accumulated. ② depicts a no-consensus case, where each expert contributes confidence to their top predictions.

If the observed label $\tilde{y}$ appears in expert $m$'s top-$K$ predictions, the expert is treated as reliable and its cumulative confidence reinforces $\tilde{y}$. Otherwise, only its top-$K$ predictions are used to avoid reinforcing a potentially noisy label.

**Noisy Label Correction and Long-Tail Calibration.** In the dynamic accumulation process, the dynamically rectified label of sample $x$ is obtained from the accumulated class frequency distribution in each round and is used for model supervision. Let $y^{r,(e)}$ be the corrected label at epoch $e$:

$$y^{r,(e)} = \arg\max_c \mathbf{F}_c^{(e)}. \tag{7}$$

Based on the rectified labels, we can recount the distribution of labels $n_c^{r,(e)}$ for class $c$ that are relatively closer to the ground-truth label distribution:

$$n_c^{r,(e)} = \sum_{i=1}^{N} \mathbb{I}(y^{r,(e)} = c), \quad c = 1, 2, \ldots, C, \tag{8}$$

where $\mathbb{I}$ is the indicator function. The rectified label distribution can be used in conjunction with various long-tail learning methods, for example, the logit adjustment (LA) loss (Menon et al., 2021) for optimization:

$$\mathcal{L}_{\text{LA}}\left(x; y^{r,(e)} = c\right) = -\log \frac{\exp\left(z_c^{IE} + \log \widehat{\mathrm{P}}(y^{r,(e)} = c)\right)}{\sum_{j=1}^{C} \exp(z_j^{IE} + \log \widehat{\mathrm{P}}(y^{r,(e)} = j))}, \tag{9}$$

where $\widehat{\mathrm{P}}(y^{r,(e)} = c)$ denotes the rectified class prior probability that is derived based on Equation (8).

The label correction for a single input image in CARE, which is performed jointly during training, is illustrated in Figure 2. The overall training algorithm is summarized in Algorithm 1 in Appendix B.

## 2.5 RATIONALE ANALYSIS

**Noise Suppression via Expert Consensus.** Multi-expert Top-$K$ consensus naturally suppresses noisy labels by leveraging agreement among experts. Since the true labels are more likely to consistently appear in Top-$K$ predictions across each expert, while random noisy labels are not, this mechanism acts as a robust filter, similar in spirit to ensemble-based noise suppression (Lee & Chung, 2020). Formally, we state:

**Theorem 1** (Reliability Amplification via Class-aware Expert Consensus). *Let $\mathbf{p}^{TE}$ and $\mathbf{p}^{IE}$ be conditionally independent, and assume the expected confidence on the true class $y$ is higher than any other class:*

$$\mathbb{E}[p_y^{TE}] > \mathbb{E}[p_c^{TE}], \quad \mathbb{E}[p_y^{IE}] > \mathbb{E}[p_c^{IE}], \quad \forall c \neq y. \tag{10}$$

*Then, for a class-specific Top-$K_c$ consensus set, the probability of both experts including $y$ is significantly higher than for any noisy label $c \neq y$:*

$$\Pr(y \in \mathcal{T}_{K_c}^{TE} \cap \mathcal{T}_{K_c}^{IE}) \gg \Pr(c \in \mathcal{T}_{K_c}^{TE} > \mathcal{T}_{K_c}^{IE}). \tag{11}$$

*Proof Sketch.* Since both experts assign higher expected confidence to the true label $y$ than any noisy label $c \neq y$, the chance of $y$ being jointly included in the Top-$K_c$ sets of both experts is much higher than for any incorrect class. This amplifies the reliability of true labels through consensus.

A formal proof is provided in Appendix C. Theorem 1 yields the consensus-driven denoising effect:

**Corollary 2** (Consensus-Based Label Refinement). *By aggregating the consensus votes over training, the method constructs a frequency matrix aligned with clean labels. As training progresses ( $e$ increases), correct consensus increasingly dominates $\mathbf{F}_y^{(e)}$ in Equation (4), leading to a bootstrapped denoising that iteratively improves label quality.*

The above corollary leads to the following implications.

**Remark 1.** (Implications of Consensus-Based Label Refinement).
- Independent expert consensus suppresses high-confidence noise.
- Aggregated over time, consensus acts as a robust voting that progressively corrects noisy labels.

**Class-Adaptive Top-$K$ Balances Head-Tail Confidence.** CARE adaptively adjusts the number of top-$K$ candidates per class based on estimated class frequency. Tail classes, with limited data and unstable predictions, require stricter consensus to suppress noise-induced interference. In contrast, head classes benefit from relaxed consensus due to concentrated confidence and sufficient supervision. This design introduces the following theoretical advantage.

**Proposition 3** (Tail-Class Consensus Robustness). *For any tail class $c$ with low $n_t^e$, selecting a smaller $K_t$ improves the expected consensus precision compared to a global $K$:*

$$\mathbb{E}[\text{Precision@}K_t] > \mathbb{E}[\text{Precision@}K], \quad \text{for } K > K_t. \tag{12}$$

The proof is provided in Appendix D. Theorem 3 yields the following practical implications.

**Remark 2.** (Implications of Class-Aware Consensus).
- The class-aware consensus mechanism offers theoretical robustness for tail-class prediction under noisy and imbalanced conditions.
- The adaptive $K_c$ accounts for class imbalance, more flexibility for head classes and conservative consensus for tail classes while preventing spurious head-class candidates from dominating tail-class decisions, ensuring a discriminative and class-sensitive consensus.
- As training progresses and labels are refined, $y^r$ becomes increasingly aligned with the true class distribution, reinforcing supervision and long-tailed bias calibration, especially for rare categories.

## 3 EXPERIMENT

### 3.1 EXPERIMENTAL SETUP AND BASELINE METHODS

**Datasets.** *Synthetic Datasets:* We evaluate CARE on both synthetic and real-world long-tailed noisy label datasets following Lu et al. (2023) and Zhang et al. (2023a). For the synthetic dataset, we construct a long-tailed noisy-label version of CIFAR-100 (Krizhevsky et al., 2009) (CIFAR-100-LTN) by subsampling its training set to follow an exponential class distribution with a predefined imbalance factor ($IF$), and then injecting symmetric, asymmetric, or joint noise at a predefined noise rate ($NR$), following the protocol in Zhang et al. (2023b). *Real-world Noise:* The WebVision-50 (Li et al., 2017) with inherent long-tailed distributions and noisy labels is adopted as real-world noise. Following SSBL$_2$ (Zhang et al., 2024), we subsample Food101N (Lee et al., 2018) into three different long-tailed noisy datasets by the $IF$ of 20, 50, and 100 with inherent noise as real-world noise. For mini-ImageNet (Jiang et al., 2020), we create a long-tailed version via exponential subsampling and inject real-world web noise, forming ImageNet-LTN$^r$ (abbreviated as Img-LTN$^r$).

**Implementation Details.** We adopt CLIP (Radford et al., 2021) ViT-B/16 as the backbone and Adaptformer (Chen et al., 2022) as the fine-tuning strategy. The model is trained for 20 epochs on all datasets with a batch size of 128 using SGD (initial learning rate 0.01, momentum 0.9, weight decay $5 \times 10^{-4}$). The computer resources for the experiments are provided in Appendix O.

**Compared Methods.** We compare CARE with three groups of methods: *Long-tail (LT) learning*: LDAM (Cao et al., 2019), NCM (Kang et al., 2020), MiSLAS (Liu et al., 2022), LA (Menon et al., 2021), and influence-balanced loss (IB) (Park et al., 2021). *Noisy label (NL) learning*: Co-teaching (CT) (Han et al., 2018), MentorNet (Lu et al., 2018), CDR (Xia et al., 2020), ELR+ (Liu et al., 2020), MoPro (Li et al., 2021), NGC (Wu et al., 2021), Sel-CL (Li et al., 2022d) DivideMix (Li et al.,

Table 2: Top-1 acc. (%) on CIFAR-100-LTN with joint noise. Res32 and Res18 are abbreviations for ResNet-32 and PreAct ResNet18, respectively.

| Dataset | CIFAR-100-LTN | | | | | | | | | |
|---|---|---|---|---|---|---|---|---|---|---|
| Imbalance Factor | 10 | | | | | 100 | | | | |
| Noise Ratio | 0.1 | 0.2 | 0.3 | 0.4 | 0.5 | 0.1 | 0.2 | 0.3 | 0.4 | 0.5 |
| CE | 48.5 | 43.3 | 37.4 | 32.9 | 26.2 | 31.8 | 26.2 | 21.8 | 17.9 | 14.2 |
| LDAM-DRW (Cao et al., 2019) | 54.0 | 50.4 | 45.1 | 39.4 | 32.2 | 37.2 | 32.3 | 27.6 | 21.2 | 15.2 |
| NCM (Kang et al., 2020) | 50.8 | 45.2 | 41.3 | 35.4 | 29.3 | 34.9 | 29.5 | 24.7 | 21.8 | 16.8 |
| MiSLAS(Liu et al., 2022) | 57.7 | 53.7 | 50.0 | 46.1 | 40.6 | 41.0 | 37.4 | 32.8 | 27.0 | 21.8 |
| Co-teaching (Han et al., 2018) | 45.6 | 41.3 | 36.1 | 32.1 | 25.3 | 30.6 | 25.7 | 22.0 | 16.2 | 13.5 |
| CDR (Xia et al., 2020) | 47.0 | 40.6 | 35.4 | 30.9 | 24.9 | 27.2 | 25.5 | 22.0 | 17.3 | 13.6 |
| Sel-CL+ (Li et al., 2022d) | 55.7 | 53.5 | 50.9 | 47.6 | 44.9 | 37.5 | 36.8 | 35.1 | 32.0 | 28.6 |
| RoLT (Wei et al., 2021) | 54.1 | 51.0 | 47.4 | 44.6 | 38.6 | 35.2 | 31.0 | 27.6 | 24.7 | 20.1 |
| RoLT-DRW (Wei et al., 2021) | 55.4 | 52.4 | 49.3 | 46.3 | 40.9 | 37.6 | 32.7 | 30.2 | 26.6 | 21.1 |
| HAR-DRW(Yi et al., 2022) | 51.0 | 46.2 | 41.2 | 37.4 | 31.3 | 33.2 | 26.3 | 22.6 | 19.0 | 14.8 |
| RCAL (Zhang et al., 2023a) | 57.5 | 54.9 | 51.7 | 48.9 | 44.4 | 41.7 | 39.9 | 36.6 | 33.4 | 30.3 |
| ECBS-Res32 (Li et al., 2024c) | 58.4 | 55.6 | 53.8 | 52.8 | 51.2 | 41.6 | 40.7 | 38.5 | 37.1 | 35.5 |
| ECBS-Res18 (Li et al., 2024c) | 63.6 | 62.3 | 60.1 | 58.2 | 55.3 | 45.6 | 44.8 | 43.0 | 39.9 | 39.1 |
| CLIP (zero-shot) | 64.4 | 64.4 | 64.4 | 64.4 | 64.4 | 64.4 | 64.4 | 64.4 | 64.4 | 64.4 |
| CLIP+CE | 82.1 | 81.3 | 80.1 | 78.8 | 77.2 | 71.0 | 69.2 | 67.8 | 66.5 | 63.8 |
| CLIP+CE w. CARE (ours) | 82.4 | 81.7 | 81.0 | 79.8 | 78.6 | 71.5 | 70.5 | 69.4 | 67.9 | 65.4 |
| CLIP+LA (Shi et al., 2024) | **84.0** | 83.0 | 82.4 | 80.8 | 79.5 | 80.5 | 79.1 | 77.4 | 76.6 | 75.3 |
| CLIP+RLA* (Chen et al., 2025) | 83.9 | 83.0 | 82.5 | 81.4 | **80.7** | 74.0 | 72.3 | 71.0 | 69.2 | 66.7 |
| CLIP+LA w. CARE (ours) | 83.9 | **83.5** | **83.0** | **81.9** | **80.7** | **80.7** | **79.8** | **78.5** | **77.5** | **76.7** |

2020), and UNICON (Karim et al., 2022). *Long-tailed noisy label (LTNL) learning*: MW-Net (Shu et al., 2019), ROLT (Wei et al., 2021), HAR (Yi et al., 2022), ULC (Huang et al., 2022), H2E (Yi et al., 2022), TABASCO (Lu et al., 2023), RCAL (Zhang et al., 2023a), ECBS (Li et al., 2024c), SSBL$_2$ (Zhang et al., 2024) and RLA (Chen et al., 2025).

## 3.2 MAIN COMPARISON RESULTS[2]

**Comparison Results on Synthetic LTNL Datasets.** We evaluate CARE on CIFAR-100-LTN under joint, symmetric, and asymmetric noise (see Tables 2 and 3). Standard LT or NL learning methods outperform CE, as they address class imbalance or label noise, respectively. LT methods benefit tail class learning by leveraging estimated class priors from observed labels, while NL methods mitigate noise through instance selection or loss correction. LTNL methods, tailored to address both challenges, achieve better overall results. Recent methods like RCAL, TABASCO, and RLA yield robust performance, but degrade under noise rates and severe class imbalance.

For example, RCAL drops to 30.3% under joint noise with $IF = 100$ and $NR = 50\%$, while TABASCO reaches only 50.5% under asymmetric noise ($IF = 10$, $NR = 40\%$). In contrast, CARE consistently outperforms both LT/LN methods and prior LTNL methods, especially under challenging settings. For example, on CIFAR-100-LTN, our method achieves 76.7% under joint noise ($IF = 100$, $NR = 50\%$) and 79.2% under symmetric noise ($IF = 10$, $NR = 60\%$), surpassing CLIP+LA by 1.4% and 3.2%, respectively. Compared to CLIP+RLA, CARE further improves accuracy by 10.0% and 2.2% under the same settings. It is also worth noting that while RLA is designed to enhance LA in noisy and imbalanced scenarios, it may occasionally underperform LA, particularly under severe noise conditions, possibly due to

Table 3: Top-1 Acc. (%) on CIFAR-100-LTN with $IF = 10$ under symmetric and asymmetric noise.

| Dataset | CIFAR-100-LTN | | | |
|---|---|---|---|---|
| Noise Type | Symmetric | | Asymmetric | |
| Noise Ratio | 0.4 | 0.6 | 0.2 | 0.4 |
| CE | 34.5 | 23.6 | 44.5 | 32.1 |
| LDAM (Cao et al., 2019) | 31.3 | 23.1 | 40.1 | 33.3 |
| LA (Menon et al., 2021) | 29.1 | 23.2 | 39.3 | 28.5 |
| IB (Park et al., 2021) | 32.4 | 25.8 | 45.0 | 35.3 |
| DivdeMix (Li et al., 2020) | 54.7 | 45.0 | 58.1 | 42.0 |
| UNICON (Karim et al., 2022) | 52.3 | 45.9 | 56.0 | 44.7 |
| MW-Net (Shu et al., 2019) | 32.0 | 21.7 | 42.5 | 30.4 |
| RoLT (Wei et al., 2021) | 42.0 | 32.6 | 48.2 | 39.3 |
| HAR (Yi et al., 2022) | 38.2 | 26.1 | 48.5 | 33.2 |
| ULC (Huang et al., 2022) | 54.9 | 44.7 | 54.5 | 43.2 |
| TABASCO (Lu et al., 2023) | 56.5 | 46.0 | 59.4 | 50.5 |
| ECBS (Li et al., 2024c) | 56.7 | 48.1 | 60.5 | 52.1 |
| CLIP zero-shot | 64.4 | 64.4 | 64.4 | 64.4 |
| CLIP+CE | 78.8 | 74.5 | 79.3 | 64.4 |
| CLIP+CE w. CARE (ours) | 80.3 | 77.2 | 80.3 | 67.5 |
| CLIP+LA (Shi et al., 2024) | 80.5 | 76.0 | 80.1 | 68.2 |
| CLIP+RLA* (Chen et al., 2025) | 80.9 | 77.0 | 80.6 | 69.3 |
| CLIP+LA w. CARE (ours) | **81.5** | **79.2** | **80.8** | **70.5** |

---

[2]Entries marked with * use our CLIP-based re-implementation; best results are in **bold**.

Table 4: Top-1 Acc. (%) on Img-LTN$^r$ with NR of 0.4.

| Dataset | Img-LTN$^r$ | |
|---|---|---|
| Imbalance Factor | 10 | 100 |
| CE | 31.5 | 31.5 |
| DivdeMix | 49.0 | 34.7 |
| UNICON | 41.6 | 31.1 |
| MW-Net | 40.3 | 31.1 |
| RoLT | 24.2 | 16.9 |
| HAR | 38.7 | 31.3 |
| ULC | 47.1 | 34.8 |
| TABASCO | 49.7 | 37.1 |
| ECBS | 50.8 | 36.9 |
| CLIP zero-shot | 54.0 | 54.0 |
| CLIP+CE | 83.4 | 80.5 |
| CLIP+CE w. CARE | 84.6 | 81.5 |
| CLIP+LA | 83.7 | 80.8 |
| CLIP+RLA* | 84.0 | 79.9 |
| CLIP+LA w. CARE | 84.4 | 81.9 |

Table 5: Top-1 Acc. (%) on WebVision-50 (WV50).

| Train | Webvision-50 | |
|---|---|---|
| Test | WV50 | IMG12 |
| CE | 62.5 | 58.5 |
| Co-teaching | 63.6 | 61.5 |
| MentorNet | 63.0 | 57.8 |
| ELR+ | 77.8 | 70.3 |
| MoPro | 77.6 | 76.3 |
| NGC | 79.2 | 74.4 |
| Sel-CL+ | 80.0 | 76.8 |
| RCAL+ | 79.6 | 76.3 |
| ECBS | 80.0 | 76.1 |
| CLIP zero-shot | 62.0 | 62.0 |
| CLIP+CE | 84.2 | 83.5 |
| CLIP+CE w. CARE | 84.7 | 83.6 |
| CLIP+LA | 85.1 | 84.3 |
| CLIP+RLA* | 85.0 | 84.2 |
| CLIP+LA w. CARE | 85.3 | 83.8 |

Table 6: Top-1 Acc. (%) on Food101N with raw noise.

| Dataset | Food101N | | |
|---|---|---|---|
| Imbalance Factor | 20 | 50 | 100 |
| LA | 64.2 | 48.7 | 46.4 |
| ELR+ | 58.6 | 49.0 | 44.1 |
| DivideMix | 64.0 | 54.7 | 52.6 |
| DivideMix-LA | 71.5 | 63.7 | 59.0 |
| DivideMix-DRW | 64.2 | 54.0 | 53.7 |
| MW-Net | 56.7 | 47.4 | 41.9 |
| HAR-DRW | 54.1 | 48.3 | 41.8 |
| H2E | 70.4 | 63.7 | 58.7 |
| SSBL$_2$ | 74.4 | 69.1 | 63.7 |
| CLIP zero-shot | 69.0 | 69.0 | 69.0 |
| CLIP+CE | 81.3 | 76.9 | 72.6 |
| CLIP+CE w. CARE | 81.6 | 79.0 | 75.2 |
| CLIP+LA | 84.8 | 84.4 | 83.7 |
| CLIP+RLA* | 84.1 | 82.2 | 77.2 |
| CLIP+LA w. CARE | 85.3 | 84.6 | 84.1 |

Figure 3: Noise rate dynamics during training of different class splits.

Table 7: Ablation study on CIFAR-100-LTN

| BE | TE | IE | NR (%) | ACC (%) |
|---|---|---|---|---|
| ✓ | | | 50.0 | 78.7 |
| ✓ | ✓ | | 50.0 | 78.7 |
| ✓ | | ✓ | 50.0 | 78.7 |
| ✓ | ✓ | ✓ | 27.8 | 80.3 |

Table 8: Comparison between CARE and BE

| | | Head | Med. | Tail | All |
|---|---|---|---|---|---|
| BE | NR ↓ (%) | 36.0 | 56.9 | 72.8 | 50.0 |
| CARE | NR ↓ (%) | 18.1 | 33.3 | 53.3 | 27.8 |
| BE | ACC ↑ (%) | 83.4 | 79.1 | 72.3 | 78.7 |
| CARE | ACC ↑ (%) | 81.8 | 81.1 | 77.3 | 80.3 |

class-agnostic label refurbishment. For a more detailed comparison, Appendix E further examines how CARE improves the performance of existing ResNet backbones.

**Comparison Results on Real-world Datasets.** Table 4, Table 5, and Table 6 report the comparative results on three real-world datasets. As shown in Table 4, our method achieves the highest top-1 accuracy on Img-LTN$^r$ under both moderate ($IF = 10$) and severe ($IF = 100$) imbalance. Notably, CARE consistently improves performance when integrated with CE or LA, surpassing existing methods such as DivideMix, ECBS, and CLIP+RLA. Besides, Appendix F analyses why CE loss outperforms the LA loss in an imbalance factor of 10. In Table 5, CARE achieves the highest accuracy (85.3%) on the WebVision test set and maintains competitive performance on ImageNet, demonstrating strong robustness to real-world noise and large-scale data. On Food101N (Table 6), our method achieves superior accuracy across all $IF$. In particular, it outperforms CLIP+RLA by a large margin under severe imbalance (e.g., +6.9% at $IF = 100$) and further improves upon CLIP+LA by up to +0.4%, indicating its complementary effect in refining label adjustment in noisy long-tailed scenarios. These results verify the effectiveness of CARE in enhancing noise tolerance and representation learning, especially under realistic and challenging long-tailed distributions.

## 3.3 FURTHER ANALYSIS

**Evolution of Class-Level Noise During Training.** As illustrated in Figure 3, the class-level noise rates on the CIFAR-100-LTN dataset (50% joint noise, $IF = 10$) decrease markedly as training progresses. After the second epoch, both overall and noise rates across classes of varying sample sizes drop sharply and then stabilize, offering increasingly reliable supervision. This trend provides increasingly accurate supervision for model training. Remarkably, despite having fewer clean samples, tail classes benefit greatly from CARE and achieve noise rates comparable to head classes, showing its effectiveness in correcting noisy labels even under severe imbalance.

**Effectiveness of Expert Collaboration.** The ablation results in Table 7 verify the individual contributions of each component (BE, TE, IE), where full integration yields a substantial performance gain (+1.6%) and significantly lowers the noise ratio from 50.0% to 27.8%. Removing either TE or IE leads to no improvement in accuracy or noise reduction, indicating that collaborative consensus among the experts is crucial. In our framework, CARE is integrated with a logit-adjustment-based

long-tailed calibration loss function. Once the labels are corrected, the improved label quality makes the calibration more precise, which can further amplify this effect on head classes. Further breakdown in Table 8 shows that CARE not only reduces noise more effectively across all class splits (especially tail: from 72.8% to 53.3%), but also achieves higher accuracy for tail classes (+5.0%) compared to the baseline expert (BE). These results validate the design of CARE and its ability to enhance both label quality and generalization in long-tailed noisy settings. Interestingly, while CARE improves tail performance, head class accuracy slightly decreases. This trade-off arises from the stronger regularization effect imposed by LA on head classes, which is further reinforced by the improved label reliability of tail classes under the proposed consensus mechanism. Appendix G-N provides more complementary analysis for CARE.

## 4 RELATED WORK

**Noisy Label Learning** Noisy label learning can generally be categorized into three groups: noisy sample separation, label correction, and noise regularization. To separate noisy samples, various metrics have been explored, including loss values (Lu et al., 2018; Han et al., 2018; Li et al., 2020), logit values (Pleiss et al., 2020), Jensen-Shannon divergence (Yao et al., 2021; Karim et al., 2022), and the distance between sample features and class prototypes (Ji et al., 2024). For label correction, many methods aim to revise noisy annotations. For example, partial label learning (Sheng et al., 2024a; Wang et al., 2022; Feng & An, 2019) assumes each training instance is associated with a candidate label set that contains the true label. PLS (Albert et al., 2023) further incorporates the confidence of the corrected labels. Other methods estimate the noise transition matrix (Yao et al., 2020; Cheng et al., 2022; Lin et al., 2024) to infer the label most likely to be correct. In terms of noise regularization, several strategies have been proposed to reduce the influence of noisy labels. These include re-weighting training examples (Ren et al., 2018; Liu & Tao, 2015) and designing noise-robust loss functions, such as backward and forward loss correction (Patrini et al., 2017), gold loss correction (Hendrycks et al., 2018), MW-Net (Shu et al., 2019), and Dual-T (Yao et al., 2020).

**Long-Tailed Noisy Label Learning** Several recent studies have been proposed to address the compounded challenges arising from the coexistence of noisy labels and unbalanced/long-tailed data. A common strategy is to identify and separate clean and noisy samples. For example, CNLCU (Xia et al., 2022) enhances the small loss method (Han et al., 2018) by designating a subset of samples with large losses as clean. TABASCO (Lu et al., 2023) addresses a complex scenario where noisy labels may cause an intrinsic tail class to appear as a head class, proposing a bi-dimensional separation metric to adapt to different cases. $SSBL_2$ (Zhang et al., 2024) separates noisy samples based on model prediction confidence and loss values. Another line of work focuses on emphasizing the importance of different samples by reweighting or regularization (Ren et al., 2018; Shu et al., 2019; Cao et al., 2021; Wei et al., 2022; Jiang et al., 2022; Chen et al., 2025). Meanwhile, improving representation learning (Zhou et al., 2022; Yi et al., 2022; Zhang et al., 2023a; Li et al., 2024c) explores intrinsic feature-space structures to mitigate noise propagation. Despite their merits, these methods underutilize the observed labels that implicitly contain weak but valuable information about the underlying ground-truth. Additional discussion of long-tailed learning is provided in Appendix J.

## 5 CONCLUDING REMARKS

In this paper, we have demonstrated that simply rectifying noisy labels in a class-agnostic way does not guarantee improved learning for tail classes, nor does it necessarily enhance overall model performance. The challenge of correcting noisy labels in tail classes is even more pronounced due to their inherent sparsity and underrepresentation. To address this, we propose the Class-Adaptive Rectification with Experts (CARE) framework, which not only rectifies noisy labels in head classes effectively but also focuses on the challenging task of correcting noisy labels in tail classes through a class-aware expert-consensus strategy. This approach ensures more accurate label rectification across all classes in various long-tailed noise scenarios, enabling the model to better capture the representative features of each class and, ultimately, improving overall performance.

While CARE achieves consistent improvements, it requires maintaining class frequency distributions during training, introducing extra memory overhead. Our future work will focus on optimizing memory usage to reduce this cost while preserving efficiency. In addition, CARE has the potential to integrate additional expert sources, such as image-aware experts from contrastive learning, to further enhance label rectification and model performance. We also leave this as an avenue for future work.

ETHICS STATEMENT

Our research focuses on enhancing the robustness of long-tailed noisy label learning with the proposed framework, unlocking the significant practical potential of learning under noisy and imbalanced data distributions. This work is both technical and practical, with direct applications to robust visual recognition in long-tailed noisy label scenarios. We have carefully considered the potential societal impacts and do not foresee any direct, immediate, or negative consequences. We are committed to the ethical dissemination of our findings and encourage their responsible use.

REPRODUCIBILITY STATEMENT

All results in this work are reproducible. We provide an anonymous link in the abstract, which contains all the code necessary for reproduction, along with the requirements needed to set up the environment. The experimental settings are detailed in Section 3.1, including implementation details such as the dataset setup and model configuration. Information on the computational resources is provided in Appendix O.

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

# APPENDIX: CLASS-AWARE RECTIFICATION WITH EXPERTS FOR ROBUST LONG-TAILED NOISY LABEL LEARNING

## A  DETAILED PROBLEM ANALYSIS FOR LONG-TAILED NOISY-LABEL LEARNING

In long-tailed noisy-label learning, performance degradation on tail classes arises from the compounding effects of *intrinsic uncertainty* due to sample scarcity and *extrinsic uncertainty* caused by label noise. We denote the observed label $\tilde{y}$ as a noisy version of the ground-truth label $y$, sampled from a corruption distribution $\eta(y \mid \tilde{y})$. Due to long-tailed distribution, the number of samples per class $n_c$ varies greatly, with $n_c \ll n_h$ for tail ($c \in \mathcal{T}$) and head ($h \in \mathcal{H}$) classes, respectively.

We summarize the core challenges as follows:

**Class-Agnostic Label Correction.** Take the label correction methods (Han et al., 2018; Li et al., 2020; Lienen & Hüllermeier, 2024; Baik et al., 2024) as an example, noise-robust methods rely on a global confidence threshold $\tau$:

$$\mathcal{D}_{\text{clean}} = \{(x_i, \tilde{y}_i) \mid \max_c p(c \mid x_i) > \tau\}. \tag{13}$$

However, for tail classes ($c \in \mathcal{T}$), the estimated confidence $p(c \mid x)$ tends to be lower due to fewer samples:

$$\mathbb{E}[p(c \mid x)] \propto n_c. \tag{14}$$

Thus, using a class-agnostic $\tau$ results in the disproportionate rejection of valid tail samples.

**Confusion Between Uncertainty and Noise.** Let $\sigma_c^2 = \text{Var}[p(c \mid x)]$ denote prediction variance for class $c$. Tail classes often have higher $\sigma_c^2$, yet this uncertainty is mistaken as noise:

$$\text{Uncertainty}(x) = 1 - \max_c p(c \mid x). \tag{15}$$

High Uncertainty$(x)$ does not always imply label corruption for tail samples. This leads to erroneous filtering:

$$\Pr[\tilde{y}_i = y_i \mid \text{Uncertainty}(x_i) > \delta] \gg 0, \quad \text{for } c = \tilde{y}_i \in \mathcal{T}. \tag{16}$$

**Amplified Mislabeling Impact via Rebalancing.** Reweighting or logit adjustment methods amplify loss from under-represented classes:

$$\mathcal{L}(x_i, \tilde{y}_i) = \alpha_c \cdot \ell(f(x_i), \tilde{y}_i), \quad \alpha_c \propto \frac{1}{n_c}. \tag{17}$$

However, since noisy label rate $\rho_c$ is typically higher for tail classes:

$$\rho_c = \Pr(\tilde{y} \neq y \mid y = c), \quad \rho_c \uparrow \text{ as } n_c \downarrow, \tag{18}$$

which leads to over-calibration driven by incorrect labels from tail classes.

**Summary.** Due to the imbalance in confidence, variance, and noise rate, class-agnostic correction and rebalancing strategies result in:

1) Over-filtering of clean tail samples,
2) Retention of confident noisy head labels,
3) Amplification of noise-induced gradient bias in tail classes.

To address this, our proposed CARE computes a class-specific confidence consensus that leverages semantic priors and adjusts to category frequency, thereby disentangling class uncertainty from annotation noise and enabling effective distribution refinement.

## B  ALGORITHM FOR CARE

The training procedure for CARE is outlined in Algorithm 1.

---

**Algorithm 1** Class-Aware Rectification with Experts

---

**Require:** Training set $\mathcal{D} = \{x_i, \tilde{y}_i\}$, pre-trained model
**Ensure:** Fine-turned model
1: Initialize $\{y_i^{r,(0)}\}_{i=1}^N \leftarrow \{\tilde{y}_i\}_{i=1}^N$, candidate count vector $\mathbf{F} \in \mathbb{R}^{N \times C} \leftarrow \{\mathbf{p}_i^{BE}\}_{i=1}^N$, class frequency $n_c^{r,(0)} = \sum_{i=1}^N \delta(\tilde{y}_i = c)$, fine-turning module $\phi$.
2: **for** $e = 1$ to $E$ **do**
3:     $K_c \leftarrow f(n_c^e)$, where $f(*)$ is monotonically increasing function , for example, $f(x) = \lfloor (x)^{1/4} \rfloor$.
4:     $\mathcal{T}_i^{TE,(e)} \leftarrow \text{TopK}(\mathbf{p}_i^{\text{IE}}|K_c), \mathcal{T}_i^{IE,(e)} \leftarrow \text{TopK}(\mathbf{p}_i^{\text{TE}}|K_c)$
5:     Update $\mathbf{F}_i^{(e)}$ for $x_i$ according to **??**
6:     $y_i^{r,(e)} \leftarrow \arg\max_c \mathbf{F}_i^{(e)}$
7:     Update $n_c^{r,(e)}$ according to Equation (8)
8:     Compute the final loss $\mathcal{L}$ using $\{x_i, y_i^{r,(e)}\}$ and $\{n_c^{r,(e)}\}_{c=1}^C$.
9:     Update $\phi$ by $\phi^{e+1} = \phi^e - \alpha \cdot \nabla \mathcal{L}$
10: **end for**

---

## C    PROOF OF THEOREM 1

*Proof.* We decompose the probability of consensus on class $y$ as follows:

$$\Pr(y \in \mathcal{T}_{K_c}^{\text{TE}} \cap \mathcal{T}_{K_c}^{\text{IE}}) = \Pr(y \in \mathcal{T}_{K_c}^{\text{TE}}) \cdot \Pr(y \in \mathcal{T}_{K_c}^{\text{IE}}). \tag{19}$$

Assuming $\mathbf{p}^{\text{TE}}$ and $\mathbf{p}^{\text{IE}}$ are conditionally independent. Under the assumption that $y$ has the highest expected confidence:

$$\mathbb{E}[\text{rank}(y)] < \mathbb{E}[\text{rank}(c)], \quad \forall c \neq y. \tag{20}$$

Since $K_c \propto n_c^e$, frequent (head) classes are allowed higher $K_c$, which increases:

$$\Pr(y \in \mathcal{T}_{K_c}^*) = \sum_{k=1}^{K_c} \Pr[\text{rank}(y) = k]. \tag{21}$$

Top-$K$ inclusion probability increases with $K_c$. For a noisy label $c \neq y$, the probability that both experts place it in Top-$K_c$ is:

$$\Pr(c \in \mathcal{T}_{K_c}^{\text{TE}}) \cdot \Pr(c \in \mathcal{T}_{K_c}^{\text{IE}}) \ll \Pr(y \in \mathcal{T}_{K_c}^{\text{TE}}) \cdot \Pr(y \in \mathcal{T}_{K_c}^{\text{IE}}), \tag{22}$$

as $c$ is not favored in expectation. Consequently,

$$\Pr(y \in \mathcal{T}_{K_c}^{\text{TE}} \cap \mathcal{T}_{K_c}^{\text{IE}}) \gg \Pr(c \in \mathcal{T}_{K_c}^{\text{TE}} \cap \mathcal{T}_{K_c}^{\text{IE}}). \tag{23}$$

$\square$

## D    PROOF OF PROPOSITION 3

*Proof.* We aim to show that for any tail class $t$ with low sample frequency $n_t^e$ at epoch $e$, using a smaller class-specific Top-$K$ value $K_t$ yields higher consensus precision compared to using a larger $K$.

**Notation**

- $\mathcal{D}_t$: the set of all samples belonging to tail class $t$.

- $E$: the set of experts.

- $\mathcal{R}_m(x, K)$: Top-$K$ classes predicted by expert $m$ for input $x$, i.e.,

$$\mathcal{R}_m(x, K) := \text{TopK}(\mathbf{z}_m).$$

**Define**

- $\mathcal{P}_t(K)$: Total number of times class $t$ appears in the Top-$K$ predictions across all experts and all samples.

- $\mathcal{T}_t(K)$: Total number of correct Top-$K$ predictions for class $t$, i.e., predictions on samples from $\mathcal{D}_t$ where $t \in \mathcal{R}_m(x, K)$.

Then the **consensus precision** for class $t$ under Top-$K$ is defined as:

$$\text{ConsensusPrecision}_t(K) := \frac{\mathcal{T}_t(K)}{\mathcal{P}_t(K)}.$$

**1. Larger $K$ increases false positives for tail class $t$.**

Let $n_t^e = |\mathcal{D}_t|$ denote the number of samples of class $t$ at epoch $e$. Then,

$$\mathcal{T}_t(K) = \sum_{x \in \mathcal{D}_t} \sum_{m \in E} \mathbf{1}[t \in \mathcal{R}_m(x, K)].$$

For each sample from class $t$, at most $|E|$ experts can correctly include $t$ in their Top-$K$, as a result:

$$\mathcal{T}_t(K) \le |E| \cdot |\mathcal{D}_t|.$$

This term is upper-bounded and cannot increase indefinitely. For $\mathcal{P}_t(K)$,

$$\mathcal{P}_t(K) = \sum_{x \in \mathcal{D}_t} \sum_{m \in E} \mathbf{1}\big[t \in \mathcal{R}_m(x, K)\big]$$

$$= \sum_{x \in \mathcal{D}_t} \sum_{m \in E} \mathbf{1}[t \in \mathcal{R}_m(x, K)] + \sum_{x \notin \mathcal{D}_t} \sum_{m \in E} \mathbf{1}[t \in \mathcal{R}_m(x, K)]$$

$$= \mathcal{T}_t(K) + \sum_{x \notin \mathcal{D}_t} \sum_{m \in E} \mathbf{1}[t \in \mathcal{R}_m(x, K)].$$

The second term accounts for false positives contributed by non-$t$ samples. For a tail class $t$, the number of non-$t$ samples is much larger than that of $t$-samples, i.e., $|\mathcal{D} \setminus \mathcal{D}_t| \gg |\mathcal{D}_t|$. The probability that $t$ appears in the Top-$K$ of a non-$t$ sample increases with $K$. Consequently, as $K$ grows, the contribution of false positives $\sum_{x \notin \mathcal{D}_t} \sum_{m \in E} \mathbb{P}[t \in \mathcal{R}_m(x, K)]$ increases, while the number of correct predictions $\mathcal{T}_t(K)$ remains bounded by $|E| \cdot n_t^e$. Then we have

$$\frac{d\mathcal{P}_t(K)}{dK} = \sum_{x \notin \mathcal{D}_t} \sum_{m \in E} \frac{\mathbf{1}[t \in \mathcal{R}_m(x, K)]}{dK} > \frac{d\mathcal{T}_t(K)}{dK} = 0,$$

Therefore, for tail classes with small $n_t^e$, increasing $K$ inflates $\mathcal{P}_t(K)$ faster than $\mathcal{T}_t(K)$.

**Additional Observation: Saturation of correct predictions vs. growth of false positives.**

In contrast, as $K$ increases, the number of samples from other classes that mistakenly include $t$ in their Top-$K$ grows significantly, leading to an increase in: $\mathcal{P}_t(K) - \mathcal{T}_t(K)$.

**2. Smaller $K$ suppresses the inclusion of $t$ on non-$t$ samples.**

Reducing $K$ limits the number of classes per prediction, which reduces the chance of incorrectly including rare class $t$ in the predictions for other classes. This slows the increase of $\mathcal{P}_t(K)$ while preserving most of $\mathcal{T}_t(K)$ for confident cases, thus improving:

$$\text{ConsensusPrecision}_t(K) = \frac{\mathcal{T}_t(K)}{\mathcal{P}_t(K)}.$$

**3. For tail classes, $\mathcal{P}_t(K)$ contains more noise under large $K$.**

Let $\epsilon_t(K)$ denote the expected proportion of false positives in $\mathcal{P}_t(K)$. For tail classes, as $K$ increases:

$$\epsilon_t(K) := \frac{\mathcal{P}_t(K) - \mathcal{T}_t(K)}{\mathcal{P}_t(K)} \uparrow,$$

indicating more of the Top-$K$ inclusions of $t$ are incorrect. Thus:

$$\text{ConsensusPrecision}_t(K) = 1 - \epsilon_t(K) \downarrow .$$

**Conclusion:** To mitigate noisy inclusions and preserve precision for rare classes, a smaller class-specific $K_t$ should be adopted for tail class $t$:

$$\text{ConsensusPrecision}_t(K_t) > \text{ConsensusPrecision}_t(K), \quad \text{for } K_t < K.$$

$\square$

## E  COMPARISON RESULTS ON RESNET BACKBONE

Table 9: Accuracy (%) Comparison of CARE (with ResNet + GloVe) on CIFAR100-LTN.

| Datasets/Methods | without CARE | with CARE |
|---|---|---|
| CIFAR100_IF100_NR50 (Joint Noise) | 45.4 | 47.0 |
| CIFAR100_IF10_NR40 (Symmetric Noise) | 61.0 | 63.4 |
| CIFAR100_IF10_NR50 (Joint Noise) | 58.9 | 61.1 |
| CIFAR100_IF10_NR60 (Symmetric Noise) | 52.3 | 56.2 |

Table 10: Accuracy (%) Comparison of CARE (with TABASCO) on CIFAR100-LTN

| Datasets/Methods | without CARE | with CARE |
|---|---|---|
| CIFAR100_IF10_NR20 (Asymmetric Noise) | 59.39 | 60.32 |
| CIFAR100_IF10_NR40 (symmetric Noise) | 56.52 | 57.20 |

Table 11: Accuracy Comparison (%) of CARE (with CLIP ResNet backbone).

| Datasets/Methods | without CARE | with CARE |
|---|---|---|
| CIFAR100_IF100_NR50 (Joint Noise) | ACC: 53.3 | NR: 42.4 ACC: 53.8 |
| CIFAR100_IF10_NR20 (Asymmetric Noise) | ACC: 63.7 | NR: 17.1 ACC: 64.1 |
| CIFAR100_IF10_NR40 (Asymmetric Noise) | ACC: 52.4 | NR: 38.3 ACC:53.2 |
| CIFAR100_IF10_NR60 (Symmetric Noise) | ACC: 56.5 | NR: 49.8 ACC: 57.5 |

For supplementary analysis, we evaluate the setting where the image expert is implemented with a ResNet backbone, and the text expert is based on GloVe embeddings (Pennington et al., 2014). The experimental results of ResNet + GloVe (Pennington et al., 2014) embeddings are presented as Table 9. Despite using less powerful models, CARE still surpasses the base model, demonstrating its generalization ability. CARE corrects noisy labels through a consensus mechanism that leverages the *discrepancy* among the predicted label distributions of different experts. Rather than depending on the absolute performance of IE or TE, CARE benefits from the diversity in their predictions, ensuring that even when the experts are relatively low-performing, as long as their outputs differ meaningfully from BE, the consensus mechanism remains effective. This robustness is also demonstrated in Table 4 to Table 6, where the zero-shot performance of CLIP (used as IE in our method) varies significantly, by up to 15% (54.0% vs. 69.0%) across two real-world datasets. Despite this large performance gap, CARE consistently achieves strong results on both datasets, maintaining accuracy above 80%, which highlights its ability to leverage expert disagreement rather than relying on expert strength alone.

In addition, we conduct experiments with TABASCO (Lu et al., 2023), a long-tailed noisy label learning method that uses ResNet as backbone. As shown in Table 10, CARE improves the performance of TABASCO under different noise ratios. These results indicate that the performance gains are not solely attributable to the backbone but arise from the core design of CARE, which effectively integrates complementary expert signals to perform robust label correction and tail-aware calibration.

Table 11 reports the performance of the CLIP ResNet backbone in our experiments. These improvements are achieved within just 20 training epochs on noisy label data. The effectiveness of CARE does not rely on the absolute accuracy of a single expert. Instead, it is driven by the *discrepancy*

among expert predictions. By comparing the cumulative frequency of predicted labels across different experts, CARE identifies and corrects potential noisy labels when there is sufficient disagreement with the base expert. Therefore, even when the auxiliary experts provide relatively inaccurate predictions, their diversity still provides meaningful signals for correction.

## F  ANALYSIS OF CE AND LA LOSS IN THE RESULTS ON MINI-IMAGENET

Table 12: Analysis of CE and LA loss in the results on mini-ImageNet.

| Class Group / Loss | Head | Tail |
|---|---|---|
| CE | 87.3 | 81.2 |
| LA | 81.7 | 85.2 |

In the results on mini-ImageNet with an imbalance factor of 10, CE loss outperforms the LA loss. An imbalance factor of 10 means that the sample size of the head classes is ten times that of the tail classes. However, mini-ImageNet is a relatively balanced and moderately sized dataset, and even under this level of imbalance, the tail classes still retain a reasonable number of training examples, typically in one hundred. This relatively sufficient amount of data for tail classes allows the model to generalize reasonably well across both head and tail classes when trained with the CE loss. Consequently, the disparity in classification performance between head and tail classes is not particularly severe, and the CE loss can maintain strong overall performance without introducing bias toward any specific subset of classes. Besides, LA explicitly modifies the logits during training to favor underrepresented classes by incorporating class prior distributions into the loss function. While this reweighting helps improve the performance on tail classes, it does so by effectively downscaling the contributions of head classes. In settings where the imbalance is mild and tail classes already have adequate data to learn from, this shift in focus can lead to overcompensation. As a result, the performance of head classes degrades significantly, overshadowing the marginal gains seen in tail-class accuracy. This trade-off ultimately leads to a decline in overall performance when using LA compared to CE. For the results on mini-ImageNet with an imbalance factor of 10, we provide the accuracy of head and tail classes as Table 12.

## G  COMPLEMENTARY ANALYSIS OF CARE COMPONENTS

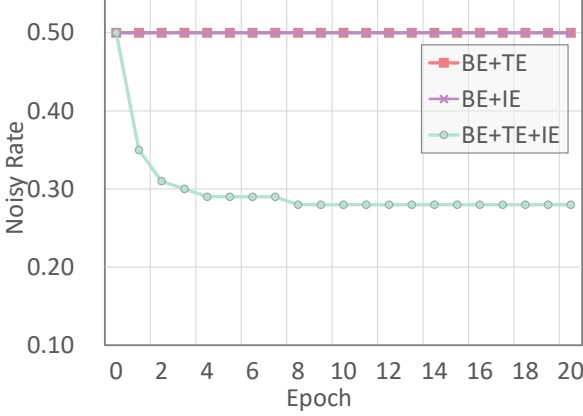

Figure 4: Noise Rate Evolution across CARE Components.

Figure 4 complements Table 7 (in Section 3.3) by illustrating how the label noise ratio evolves across epochs. It further demonstrates that a single expert alone is insufficient to alter the base expert (BE). Since the confidence scores from the image expert (IE) and text expert (TE) remain consistently below 1. Consequently, their individual contributions to the frequency accumulation vector $\mathbf{F}$ (defined in **??**)

are insufficient to surpass the accumulated support from BE. The single expert cannot accumulate enough frequency to override BE and induce label correction Collaborative voting among multiple experts for robust label refinement is necessary.

## H  ANALYSIS OF CUMULATIVE FREQUENCY-BASED CORRECTION WITH VARYING BE WEIGHT

Table 13: Effect of Expert Combinations on Noisy Label Correction (BE Weight = 0.5, Symmetric Noise). NR denotes Noise Rate after correction.

| Methods / Datasets | BE+TE | BE+IE |
|---|---|---|
| CIFAR100_IF10_NR50 | NR: 31.8 ACC: 79.8 | NR: 29.3 ACC: 79.7 |

To further clarify the core mechanism of our correction strategy, which relies on cumulative frequency comparison, we analyze the case where the BE weight is set to 1. In this setting, correcting noisy labels in BE requires the cumulative frequency of predictions from IE and TE to exceed that of the noisy labels in BE. As shown in Table 7 (in Section 3.3), neither IE nor TE alone can accumulate sufficient clean signals to achieve this correction. We then conducted additional experiments with the BE weight reduced to 0.5. The results in Table 13 demonstrate that, under this setting, either IE or TE alone is sufficient to correct noisy labels and improve model performance.

## I  ABLATION ON THE FORM OF TOP-$K$ PREDICTIONS

Table 14: Ablation Study on $K$ (CIFAR-100-LTN, IF=100, 20%, Symmetric Noise).

| $K$ | ACC (%) |
|---|---|
| CLIP+LA (baseline) | 76.0 |
| Step form (H:8 M:4 T:1) | 76.5 |
| Step form (H:8 M:3 T:2) | 76.3 |
| Step form (H:7 M:4 T:2) | 76.4 |
| Step form (H:8 M:4 T:2) | 76.7 |
| Step form (H:9 M:4 T:2) | 76.9 |
| Step form (H:8 M:5 T:2) | 76.3 |
| Step form (H:8 M:4 T:3) | 76.6 |
| Exponential form ($K_i = n_i^{0.25}$) | 77.3 |
| Logarithmic form ($K_i = \log(n_i)$) | 77.3 |
| Linear form | 77.5 |

The $K$ value for each class is proportional to the number of samples in that class. Specifically, it is computed as the one-fourth power of the number of samples in that class. We empirically adopt this power-law form as a practical choice. We also conducted ablation experiments on the calculation form of the hyperparameter $K$, and the results are shown as Table 14. "H", "T" and "T" refer to Head, Medium, and Tail classes, and $n_i$ denotes the number of samples in class i. The linear form is computed as $K_i := \frac{n_i - n_{min}}{(n_{max} - n_{min})} * (K_{max} - K_{min}) + K_{min}$. $n_{max}$ and $n_{min}$ represent the maximum and minimum sample counts, which are set to 9 and 1, respectively. The experimental results demonstrate that different forms of $K$, particularly those proportional to class frequency, yield stable and improved performance.

Global vs. Class-wise $K$ (Table 15): Using a global $K$ for all classes improves overall accuracy compared to baseline (equivalent to $K = 1$), but the class-wise $K_c$ consistently achieves higher performance, particularly benefiting tail classes.

Table 15: Comparison of class-wise $K$ and global $K$.

| Datasets $K$ | CIFAR-100-LTN ($IF = 10$, $NR = 20$, AN) | | | | Img-LTN$^r$ ($IF = 100$, $NR$=40) | | | |
|---|---|---|---|---|---|---|---|---|
| | ACC | Head | Med | Tail | ACC | Head | Med | Tail |
| Global $K = 4$ | 80.7 | 81.0 | 82.7 | 77.9 | 80.7 | 81.1 | 96.0 | 73.6 |
| Global $K = 8$ | 80.3 | 80.3 | 82.7 | 77.3 | 80.2 | 80.9 | 96.0 | 74.4 |
| Class-wise $K_c$ | 80.8 | 80.9 | 82.5 | 78.6 | 81.9 | 83.0 | 94.5 | 74.8 |

## J  RELATED WORK OF LONG-TAIL LEARNING

Long-tailed learning methods typically assume that datasets are correctly labeled (Cui et al., 2019), and adopt class-wise strategies that can be broadly categorized into four levels (Li, 2022; Shi et al., 2024). Input level: These approaches manipulate the training data to enhance learning, including re-weighting, re-sampling (Cui et al., 2019), and data augmentation techniques (Cubuk et al., 2019; 2020). Representation level: Methods at this level modify model training to decouple representation learning from classifier learning, such as in decoupled training (Kang et al., 2020; Zhong et al., 2021) and BBN-based approaches (Zhou et al., 2020; Zhang et al., 2021), where features are learned on imbalanced data, followed by classifier retraining with class-balanced or reversed samples. Ensemble strategies are also used, including redundant ensembles (Wang et al., 2021; Li et al., 2022a;b; Cai et al., 2021) and complementary ensembling (Zhou et al., 2020; Cui et al., 2023), which leverage diverse expert outputs or data partitions. Output level: These methods directly adjust model outputs to model generalization and classifier performance. Logit adjustment (Menon et al., 2021; Ren et al., 2020) re-scales predictions, while re-margining techniques (Cao et al., 2019; Li et al., 2023; Menon et al., 2021; Li et al., 2022c) assign class-dependent margins to promote better tail-class separation. Gradient level: CCSAM (Zhou et al., 2023b) and ImbSAM (Zhou et al., 2023a) adapt Sharpness-aware Minimization (Foret et al., 2021) to strengthen tail-class learning by adjusting gradient contributions based on class frequency. GNM (Li et al., 2024a) introduces parameter-independent perturbations to reduce head-class dominance during training.

## K  GENERALIZATION ANALYSIS OF CARE

Table 16: Experimental results (%) of two additional noise types.

| Datasets/Methods | without CARE | with CARE |
|---|---|---|
| CIFAR100_IF100_NR40 (Uniform Noise) | ACC: 70.8 | ACC: 74.1 |
| CIFAR100_IF100_NR60 (Uniform Noise) | ACC: 65.3 | ACC: 69.3 |
| CIFAR100_IF100_NR20 (Pair-Flip Noise) | ACC: 77.9 | ACC: 79.4 |
| CIFAR100_IF100_NR40 (Pair-Flip Noise) | ACC: 67.0 | ACC: 70.7 |

We evaluated two additional noise types across multiple datasets:

1. Standard Symmetric Noise (Uniform Noise): Its transition matrix is defined as

$$T_{ii} = 1 - \eta \quad \text{(probability of keeping the correct label)},$$

$$T_{ij} = \frac{\eta}{\text{num\_classes} - 1}, \quad i \neq j \quad \text{(probability of flipping from class } i \text{ to class } j).$$

2. Asymmetric Pair-Flip Noise: Instead of flipping to any random class, each class is flipped to one specific, visually similar class, modeling realistic inter-class confusion.

The experimental results are summarized in the Table 16. As shown, CARE consistently improves performance under both standard symmetric noise and asymmetric pair-flip noise. Specifically, for standard symmetric noise (NR40 and NR60), CARE achieves performance gains of 3.3 and 4.0 points, respectively, demonstrating its effectiveness in mitigating randomly distributed label corruption. Under the more challenging pair-flip noise (NR20 and NR40), which arises from class similarity, CARE still provides improvements of 1.5-3.7 points. Overall, these results show that CARE generalizes well across different noise types, enhancing robustness to both random and structured label noise.

# L   ANALYSIS OF MACRO F1-SCORE

Table 17: The Macro F1-score (%) across multiple datasets.

| Datasets/Methods | without CARE | with CARE |
|---|---|---|
| CIFAR100_IF100_NR30 (Joint Noise) | 77.2 | 78.3 |
| CIFAR100_IF100_NR50 (Joint Noise) | 75.0 | 76.4 |
| CIFAR100_IF10_NR40 (Symmetric Noise) | 80.3 | 81.4 |
| CIFAR100_IF10_NR60 (Symmetric Noise) | 75.7 | 78.9 |
| CIFAR100_IF10_NR20 (Asymmetric Noise) | 80.0 | 80.7 |
| CIFAR100_IF10_NR40 (Asymmetric Noise) | 67.6 | 69.7 |

Macro F1-score is the average of the F1-score computed independently for each class, giving equal weight to all classes regardless of their size. By combining precision and recall, Macro F1-score captures how well a model balances false positives and false negatives across classes. This makes it sensitive to class imbalance, as poor performance on minority classes cannot be offset by strong performance on majority classes.

We evaluated our method in terms of Macro F1-score(%) across multiple datasets, as shown in Table 17. CARE consistently improves Macro F1-score under all noise settings. For joint noise (NR30/NR50), it achieves gains of approximately 1-1.4 points. For symmetric noise (NR40/NR60), CARE achieves consistent improvements, including a substantial 3.2-point increase at $NR = 60$. Under the more challenging asymmetric noise scenario (NR20/NR40), CARE continues to demonstrate measurable performance gains. Overall, CARE reliably enhances class-balanced performance across different noise types and severities.

# M   EXPERIMENTAL RESULTS (%) OF CLOTHING1M

Table 18: Experimental results (%) of Clothing1M.

| Datasets/Methods | without CARE | with CARE |
|---|---|---|
| Clothing1M | 71.2 | 71.4 |

The experimental results of Clothing1M are presented as in Table 18. CARE yields a modest improvement on Clothing1M (71.2 vs. 71.4), which can be attributed to the dataset's abundant training samples and atypical class imbalance. Specifically, even the least frequent classes contain 19,743 samples, which is substantially higher than the tail classes in typical long-tailed datasets, indicating that these classes are already well represented. Consequently, the impact of methods designed to address tail-data scarcity is limited. Furthermore, in Clothing1M, some classes that are rare in the test set actually have a relatively large number of training samples, which is fundamentally different from real-world long-tailed distributions. In typical long-tailed scenarios Menon et al. (2021); Zhou et al. (2020); Li et al. (2024b), each class is considered equally important, and tail classes are consistently underrepresented both in training and testing, leading to a relatively balanced test set that reflects the importance of all classes. This contrasts with Clothing1M, where the training set abundance for some rare test classes reduces the effect of tail scarcity. Despite these factors, CARE still achieves a measurable gain, showing that the method remains effective even when the dataset deviates from the specific long-tailed, noisy conditions it is primarily designed for.

# N   ANALYSIS OF NOISE RATES FOR DIFFERENT CLASSES IN RLD

The noise rates for samples of different frequencies in RLD 1, 2, and 3 (as shown in Table 1) are presented in Table 19. Compared with RLD1 and RLD2, RLD3 exhibits a substantially more balanced noise-rate distribution across the head, medium, and tail categories. As shown in the table, the noise rates of RLD1 and RLD2 display a clear long-tailed pattern, where the tail classes suffer from significantly higher noise (e.g., 70.2% in RLD1 and 59.2% in RLD2), while the head classes

Table 19: Noise rates (%) for different classes in RLD 1, 2, and 3.

| Methods/Classes | Head | Med. | Tail |
|---|---|---|---|
| RLD1 | 17.4 | 21.6 | 70.2 |
| RLD2 | 17.0 | 30.5 | 59.2 |
| RLD3 | 27.9 | 27.8 | 31.2 |

remain comparatively clean. In contrast, RLD3 reduces this discrepancy: the noise rates of the three frequency groups (27.9% for head, 27.8% for medium, and 31.2% for tail) are closely aligned. This indicates that RLD3 mitigates the frequency-dependent bias of label noise, leading to a more uniform noise distribution. Such balance is beneficial for learning algorithms, as it prevents models from being disproportionately affected by high-noise tail classes.

## DISCLOSURE OF LLM USAGE

In line with the ICLR 2026 policy, we acknowledge the use of large language models (e.g., ChatGPT, DeepSeek) solely for grammar refinement and language polishing. Their use was strictly confined to enhancing the clarity and readability of the text. They were not involved in research conception, methodological design, algorithm development, data collection or analysis, experimental procedures, interpretation of results, or any component of the technical contributions. All scientific ideas and contributions presented in this work are original to the authors.

## O  COMPUTATIONAL RESOURCES

To ensure a fair comparison, all the experiments are executed on the following hardware: Intel (R) Xeon (R) Gold 5220, operating at 2.20GHz, equipped with 251GB RAM, and a single NVIDIA GeForce RTX 3090 GPU with 24 GB of memory.

