# OpenReview forum: "Class-Adaptive Rectification with Experts for Robust Long-Tailed Noisy Label Learning"
_ICLR.cc/2026/Conference — Submitted to ICLR 2026_

### Official Review · Reviewer_oswe · 2025-10-31

**Soundness:** 2
**Presentation:** 2
**Contribution:** 2
**Rating:** 2
**Confidence:** 4

**Summary:**

The paper propose a method for label correction by merging information from text-image alignment, pseudo label and noise label. Also it use class adaptive strategy to apply different level of noise correction for class of different frequency.  The experiments are conducted on both synthetic data and real data. Compared with CLIP based methods, it can bring further improvement. However, there lack ablations to support the key claims, and there missing some details for the whole pipeline. The suggest is to reject.

**Strengths:**

The proposed method merges priors in text/image alignment(TE), pseudo labels (IE) and noise labels (BE) to generate a better label closer to ground truth distribution, yielding to better results. Also it emphasizes different impact of label noise across classes.

**Weaknesses:**

The motivation sounds good but there lack experiments supporting the claim.
Why the tail classes requires more corrections. There is no analysis showing something like marginal benefit when applying label corrections for head classes.
How the proposed method apply more corrections on the tail classes? There is a theorem (Proposition 3) but the proof for it is empirical and there is no comparison for different settings for an actual proof.

See questions part below for other concerns

**Questions:**

1. How is the NR calulated in Table1. From figure 1 it seems RLD 1,2,3 have much lower noise level than reported in table 1.
2. What is NR for data at different frequency in RLD 1,2,3 in Table1?
3. For the proof of Theorem 3, the proof seems empirical but not a strict proof. Like "Since tail class t has
low sample frequency, ... Thus, Pt(K) grows faster than Tt(K). "  How low frequency class t should be to make it happen? At least Pt(K), Tt(K) should be written in a function of class frequency. Otherwise it is hard to tell whether the claim is correct
4. The relationship between Kc and class frequency n_c is the key for class aware noise controlling. How the choice of the relationship affect the results? Like using a global K for all the class, what the actual result would be?  Will the noise level of tail class increase or decrease?
5. The name of Table 7 and Table 8 should exchange?
6. In Table 7, the first three rows have exact the same NR and accuracy, which is strange.
7. Does the improvement come from the class-aware noise assignment or the expert mixing strategy? From table 1 and figure 1, the RLD 3 have more noise level at the "Many" class than RLD 1,2 but it still can achieve better accuracy at head classes.
8. In Figure 2 case1, there is a class with 0.4 probability in the IE part,  which is removed in the final accumulation. I don't find the corresponding strategy in the paper. Is that a mistake in the figure?
9. The pipeline remains unclear to me. The AdaptFormer and w is for calculation of IE, and used in label correction. After that, which part of the model will be trained using the corrected label to get the final results? How is the AdaptFormer and w initialized? Do they need cold start training using the noise label?

---

> ### Author Response · Authors · 2025-11-20
>
> >**W1**: The motivation sounds good but there lack experiments supporting the claim. Why the tail classes requires more corrections. There is no analysis showing something like marginal benefit when applying label corrections for head classes. How the proposed method apply more corrections on the tail classes? There is a theorem (Proposition 3) but the proof for it is empirical and there is no comparison for different settings for an actual proof.
>
> ---
>
> **R1**: We appreciate the reviewer's insightful comment regarding the need for more concrete evidence and justification for our core claim. We acknowledge that the empirical analysis for Proposition 3 can be presented more explicitly.
>
> While we understand the reviewer’s concern, we respectfully believe that the supporting evidence is present, though currently dispersed across the initial analysis and ablation studies. -- In the revised version, we will more clearly integrate and highlight these connections to improve the clarity and coherence of the evidence.
>
>
>
> 1. Marginal benefit for Head Classes Correction
>
> Long-tailed noisy label learning is a realistic yet highly challenging compound problem, where long-tail imbalance and label noise jointly influence the model. Prior work [R1-R4] has shown that long-tailed re-balancing methods often lead to a reduction in head-class accuracy, as these methods place greater emphasis on improving tail-class performance. In our framework, CARE is integrated with a logit-adjustment-based long-tailed calibration loss function. Once the labels are corrected, the improved label quality makes the calibration more precise, which can further amplify this effect on head classes.
>
> At the same time, learning from cleaner labels naturally enhances overall training quality. The interaction between these two factors, namely more accurate labels and stronger long-tailed calibration, exhibits the observed behavior in head classes. This is an expected and reasonable outcome in long-tailed noisy label settings.
>
> We have added this clarification and analysis to the revised version for completeness.
>
>
> Reference:
>
> [R1] Menon, et al. Long-tail learning via logit adjustment. _ICLR_, 2021.
>
> [R2] Mengke Li, et al. Long-tailed visual recognition via gaussian clouded logit adjustment. _CVPR_, 2022.
>
> [R3] Bowen Dong, et al. LPT: long-tailed prompt tuning for image classification. _ICLR_, 2023.
>
> [R4] Jiang-Xin Shi, et al. Long-tail learning with foundation model: Heavy fine-tuning hurts. _ICML_, 2024.
>
>
> 2-1. Empirical Justification for Tail Class Correction
>
> The need for more correction in tail classes is supported by our initial analysis in Section 4.1 (Empirical Observation of Long-Tailed Noisy Labels).
>
> - **Uneven Noise Impact:** As shown in Figure 1 (Right), our analysis of the *true* noise distribution reveals that tail classes (small $n_c^e$) have a significantly higher percentage of noisy labels compared to head classes (large $n_c^e$). For example, the noise rate for the smallest classes can be 2-3 times that of the largest classes.
> - **Insufficient Correction by Baselines:** Table 1 presents a preliminary study showing that class-agnostic correction methods (RLD 1 and RLD 2) fail to improve performance over the noisy baseline, demonstrating that a non-adaptive approach cannot effectively manage this uneven noise and likely falls short in the tail.
>
> This evidence strongly validates our motivation that tail classes, having higher inherent noise, require a more stringent and effective correction mechanism.
>
> 2-2. How CARE Applies More Correction to Tail Classes
>
> Our Class-Adaptive Rectification (CARE) intrinsically applies a more conservative (i.e., less permissive to noise) correction to tail classes through the class-specific parameter $K_c$.
>
> - **Mechanism:** The adaptive parameter $K_c$ (Algorithm 1) is set proportional to the class frequency $n_c^e$: $K_c \propto (n_c^e)^{1/4}$.
> - **Effect:** Since tail classes have smaller $n_c^e$, their corresponding $K_c$ are also small (often $K_c=1$). This means that for a sample to be corrected in a tail class, the consensus must be based on a very small set of top predictions.
> - Small $K_c$ for Tail Classes: The consensus is based on a very limited set of top predictions, enforcing a strict consensus requirement. Only the single most confident prediction from the experts can be considered, which makes the correction conservative and reduces the chance of mistakenly reinforcing noisy labels. This thus makes incorrect labels easier to reject.
> - Large $K_c$ for Head Classes: The consensus can draw from a wider range of top predictions, allowing for a more permissive consensus. This helps prevent unnecessary corrections in head classes, which typically have lower noise rates and sufficient training samples.
>
> This adaptive setting is the method by which we apply a disproportionately higher scrutiny (i.e., effective correction power) to the noisier tail classes.

---

> ### Author Response · Authors · 2025-11-20
>
> 3. Regarding Proposition 3
>
> A formal proof of Proposition 3 is revised in Supplementary Material (**Appendix D**), which theoretically shows that for tail classes, a smaller class-specific Top-$K$ improves consensus precision by reducing false positives.
>
> In addition, we conduct experiments under various settings, which empirically validate this theoretical finding and confirm the effectiveness of our class-aware Top-$K$ mechanism.
>
>
>
> - **Comparison to Fixed $K$ (Ablation Study):**
>
> **Table R4.** Experimental results (%) of global $K$.
> | K/Datasets       | CIFAR100_IF10_NR20 (Asymmetric Noise)                | Mini_ImageNet_IF100_NR40       |
> | - | - | - |
> | Global k: 4      | ACC: 80.7 Head: 81.0 Med: 82.7 Tail: 77.9 (NR: 16.7) | ACC: 80.7 Head: 81.1 Med: 96.0 Tail: 73.6 |
> | Global k: 8      | ACC: 80.3 Head: 80.3 Med: 82.7 Tail: 77.3 (NR: 18.5) | ACC: 80.2 Head: 80.9 Med: 96.0 Tail: 74.4 |
> | Class-wise $K_c$ | ACC: 80.8 Head: 80.9 Med: 82.5 Tail: 78.6 (NR: 16.3) | ACC: 81.9 Head: 83.0 Med: 94.5 Tail: 74.8 |
>
> **Global vs. Class-wise $K$ (Table R4)**: Using a global $K$ for all classes improves overall accuracy compared to baseline (equivalent to $K = 1$), but the class-wise $K_c$ consistently achieves higher performance, particularly benefiting tail classes.
>
> **Different Forms of $K$**: **Appendix I** (Table 14) compares the adaptive $K_c$ against various forms. This demonstrates that setting $K$ proportional to class frequency provides a consistent benefit and is more suitable than a single fixed $K$.
>
>
> - **Head/Tail Accuracy:** The gain in Tail Class Accuracy (e.g., in Table R4) achieved by CARE further demonstrates that the adaptive mechanism successfully mitigates the long-tail noise issue, which is the core implication of Proposition 3.

---

> ### Author Response · Authors · 2025-11-20
>
> >**Q1**: How is the NR calulated in Table1? From figure 1 it seems RLD 1,2,3 have much lower noise level than reported in table 1.
>
> ---
>
> **A1**: The Noise Rate (NR) in Table 1 is calculated based on the total number of noisy labels (after correction) relative to the original clean labels (before noise injection). Figure 1, however, shows per-class noise distributions, which may visually suggest lower noise levels in subsets like RLD 1, 2, and 3. The difference arises from the global NR calculation in Table 1 versus the subset-level noise visualization in Figure 1.
> The original clean labels are used solely for measuring the noise ratio and are not involved in the training process.
>
>
> >**Q2**: What is NR for data at different frequency in RLD 1,2,3 in Table1?
>
> ---
>
> **A2**: The noise rates for samples of different frequencies in RLD 1, 2, and 3 (as shown in Table 1) are presented in Table R5.
>
> **Table R5.** Noise rates for different classes in RLD 1, 2, and 3.
>
> | Methods/Classes | Head  | Med.  | Tail  |
> | - | - | - | - |
> | RLD1            | 17.4% | 21.6% | 70.2% |
> | RLD2            | 17.0% | 30.5% | 59.2% |
> | RLD3            | 27.9% | 27.8% | 31.2% |
>
> Compared with RLD1 and RLD2, RLD3 exhibits a substantially more balanced noise-rate distribution across the head, medium, and tail categories. As shown in Table R5, the noise rates of RLD1 and RLD2 display a clear long-tailed pattern, where the tail classes suffer from significantly higher noise (e.g., 70.2% in RLD1 and 59.2% in RLD2), while the head classes remain comparatively clean. In contrast, RLD3 reduces this discrepancy. The noise rates of the three frequency groups (27.9% for head, 27.8% for medium, and 31.2% for tail) are closely aligned. This demonstrates that RLD3 alleviates the frequency-dependent bias of label noise, resulting in a more uniform noise distribution. With a more consistent overall noise rate, applying stricter correction to the tail classes helps improve the overall model performance. This further validates our motivation that noise label learning in long-tailed distributions requires more rigorous correction for tail classes.
>
>
> >**Q3**: For the proof of Proposition 3, the proof seems empirical but not a strict proof. Like "Since tail class t has low sample frequency, ... Thus, Pt(K) grows faster than Tt(K). " How low frequency class t should be to make it happen? At least Pt(K), Tt(K) should be written in a function of class frequency. Otherwise it is hard to tell whether the claim is correct
>
> ---
>
> **A3**: We appreciate the reviewer’s comment. For $\mathcal{P}_t(K)$, we have:
> \begin{aligned}
> \mathcal{P}_t(K) &= \sum _{x \in \mathcal{D}_t} \sum _{m \in E} \mathbf{1}\big[t \in \mathcal{R}_m(x,K)\big]
> \+ \sum _{x \notin \mathcal{D}_t} \sum _{m \in E} \mathbf{1}\big[t \in \mathcal{R}_m(x,K)\big].
> \end{aligned}
>
> For $\mathcal{T}_t(K)$:
> $\mathcal{T}_t(K) = \sum _{x \in \mathcal{D}_t} \sum _{m \in E} \mathbf{1}\big[t \in \mathcal{R}_m(x,K)\big],$
> We have revised the argument in the supplementary material (see **Appendix D**) to clarify how the growth of $\mathcal{P}_t(K)$ and $\mathcal{T}_t(K)$ depends on the class frequency.

---

> ### Author Response · Authors · 2025-11-20
>
> >**Q4**: The relationship between Kc and class frequency n_c is the key for class aware noise controlling. How the choice of the relationship affect the results? Like using a global K for all the class, what the actual result would be? Will the noise level of tail class increase or decrease?
>
> ---
>
> **A4**: In this paper, classes with higher sample counts (head classes) are assigned larger K, while those with fewer samples (tail classes) are assigned smaller K, allowing the model to allocate trust more appropriately across the class spectrum. To investigate the impact of different proportional strategies for selecting K, we conducted an ablation study on various K values. The results are presented in Table 14 of **Appendix I**. The experimental results demonstrate that different forms of K, particularly those proportional to class frequency, yield stable and improved performance. In addition, we conducted supplementary ablation studies where K was set inversely proportional to the number of samples per class, as well as using a global fixed K value. The experimental results are presented as follows:
>
> **Table R4.** Experimental results (%) of global $K$.
> | K/Datasets       | CIFAR100_IF10_NR20 (Asymmetric Noise)                | Mini_ImageNet_IF100_NR40       |
> | - | - | - |
> | Global k: 4      | ACC: 80.7 Head: 81.0 Med: 82.7 Tail: 77.9 (NR: 16.7) | ACC: 80.7 Head: 81.1 Med: 96.0 Tail: 73.6 |
> | Global k: 8      | ACC: 80.3 Head: 80.3 Med: 82.7 Tail: 77.3 (NR: 18.5) | ACC: 80.2 Head: 80.9 Med: 96.0 Tail: 74.4 |
> | Class-wise $K_c$ | ACC: 80.8 Head: 80.9 Med: 82.5 Tail: 78.6 (NR: 16.3) | ACC: 81.9 Head: 83.0 Med: 94.5 Tail: 74.8 |
>
> Table R4 shows that using a fixed global $K$ value on the CIFAR datasets slightly reduces the noise rate and leads to a small performance improvement. However, on the Mini-ImageNet dataset, where the injected noisy labels are collected from the web and do not have corresponding clean annotations, using a global K value results in degraded performance. These findings indicate that a global K value does not provide consistent performance gains, and its optimal choice is highly dependent on the dataset, making it an unstable hyperparameter.
>
> >**Q5**: The name of Table 7 and Table 8 should exchange?
>
> ---
>
> **A5**: Thank you for pointing out the naming issue. Tables 7 and 8 can indeed be named more precisely. Originally, the title of Table 8 was intended to highlight the effect of expert consensus, but it may have caused confusion. To clarify, Table 7 could be titled “Ablation study of the three experts on the CIFAR-100-LTN dataset,” while We have updated the caption of Table 8 to "Comparison between CARE and BE" to avoid confusion, taking into account the space constraints.
>
>
> >**Q6**: In Table 7, the first three rows have exact the same NR and accuracy, which is strange.
>
> ---
>
> **A6**: To clarify, the core mechanism of our correction strategy is based on cumulative frequency comparison. Specifically, when the weight of BE is set to 1, to correct potential noisy labels in BE, the cumulative frequency of predicted labels from IE and TE must surpass the cumulative frequency of noisy labels in BE. Neither IE nor TE alone can accumulate sufficient clean signal to achieve this, as shown in Table 7. We conducted experiments with two experts where the BE weight was set to 0.5. The results are shown in Table 13 and analyzed in **Appendix H**, which show that after halving the BE weight, either IE or TE alone is capable of correcting noisy labels and improving model performance.
>
>
> >**Q7**: Does the improvement come from the class-aware noise assignment or the expert mixing strategy?
>
> ---
>
> **A7**: The improvement in model performance stems from the expert-mixed cumulative frequency strategy, which is driven by the class-specific expert consensus mechanism. This strategy aggregates frequency information from multiple experts, with each expert's contribution guided by the class-specific consensus. It ensures that high-confidence classes are given more weight in the frequency accumulation, while low-confidence classes are down-weighted. The objective is to accumulate more frequency for high-confidence classes, while minimizing the impact of low-confidence ones, thereby accurately identifying the correct class for the given image.

---

> ### Author Response · Authors · 2025-11-20
>
> >**Q8**: From table 1 and figure 1, the RLD 3 have more noise level at the "Many" class than RLD 1,2 but it still can achieve better accuracy at head classes.
>
> ---
>
> **A8**: The performance is not solely determined by the noise rate. RLD 3 intentionally uses class-aware priors in label rectification, enabling it to retain high-confidence samples in dominant classes and filter out misleading low-confidence ones. In other words, not all noisy labels have the same detrimental effect on learning. RLD 3 intentionally leverages class-aware priors during label rectification, which enables it to preserve reliable high confidence samples in dominant (Many) classes while filtering out misleading low confidence ones. Consequently, even if the apparent noise proportion is higher, the remaining labels better reflect the true class distribution and decision boundary. In contrast, the class-agnostic methods (RLD 1 and RLD 2) may reduce the overall noise ratio but tend to overcorrect or distort the feature label alignment in high frequency classes, leading to lower head class accuracy. Therefore, RLD 3 achieves better head class performance by maintaining effective labels rather than simply minimizing the numerical noise ratio.
>
>
>
> >**Q9**: In Figure 2 case 1, there is a class with 0.4 probability in the IE part, which is removed in the final accumulation. I don't find the corresponding strategy in the paper. Is that a mistake in the figure?
>
> ---
>
> **A9**: This is not an error in the figure. In Figure 2, Case 1, the class with a probability of 0.4 corresponds to the highest predicted probability from the IE model. This is compared with the top-predicted class from the TE model (also with a probability of 0.4) and the BE’s prediction. Since the top class from the IE model differs from the BE’s prediction, while the TE’s top class matches the BE’s prediction, the expert consensus mechanism deems the IE’s prediction less reliable, thus reducing its frequency and effectively removing it from consideration. In contrast, the TE’s prediction is deemed more trustworthy, and its frequency is increased. This process is reflected in Equation 4.
>
> >**Q10**: The pipeline remains unclear to me. The AdaptFormer and w is for calculation of IE, and used in label correction. After that, which part of the model will be trained using the corrected label to get the final results? How is the AdaptFormer and w initialized? Do they need cold start training using the noise label?
>
> ---
>
> **A10**: w in Figure 1 denotes the classifier, whose parameters are initialized using the class text features produced by the text encoder. The parameters of AdaptFormer are initialized to zero. Both the parameters of AdaptFormer and w are updated during training using the rectified labels. In other words, AdaptFormer and w jointly participate in the computation of IE, whose results are used for label rectification. The rectified labels are then fed back to further train AdaptFormer and w. It is worth noting that in all training rounds, AdaptFormer and w are trained using the dynamically updated rectified labels, which are refreshed at each round.

---

> ### Author Response · Authors · 2025-11-28
>
> Dear Reviewer oswe,
>
> We are grateful for your valuable suggestions that have strengthened the rigor of our paper. In consideration of the substantial workload reviewers face, we provide a condensed overview of our key responses below to assist your evaluation.
>
> - **More corrections for tail classes**: Our class-adaptive $K-c$ enforces stronger correction on tail classes by assigning smaller Top-$K$ to low-frequency classes. A smaller $K$ makes the consensus more selective, which filters out unreliable predictions and increases the proportion of corrected samples.
>
> - **NR in Table 1**: The noise rates for samples of different frequencies in RLD 1, 2, and 3 (as shown in Table 1) are presented in Table R5. Compared with RLD1 and RLD2, RLD3 exhibits a substantially more balanced noise-rate distribution across the head, medium, and tail categories.
>
> - **The proof of Proposition 3**: We further strengthen Proposition 3 with a clearer class-frequency–dependent argument in the supplementary material (see **Appendix D**).
>
> - **Global $K$**:
> Ablation with a global $K$ (Table R4) shows inconsistent gains, and its optimal value varies across datasets, making it an unstable hyperparameter.
>
> - **Clarification of Table 7**: The identical results in the first three rows of Table 7 occur because with BE weight 1, neither IE nor TE alone can surpass BE’s cumulative frequency to correct noisy labels. When the BE weight is reduced (Table 13, Appendix H), either IE or TE alone can correct labels and improve performance.
>
> - **Clarification of Figure 2 & the correction pipeline of CARE**:
> The mechanism in Figure 2 reflects our expert consensus strategy: when the IE prediction conflicts with the BE prediction but the TE prediction agrees with it, the IE result is deemed less reliable and its contribution is down-weighted, while the TE result is reinforced. This reliability-based frequency adjustment follows Equation 4 and leads to the removal of the IE prediction in Case 1.
> For the taining pipeline, the classifier $w$ is initialized from text features and AdaptFormer is initialized to zero. Both are jointly updated using dynamically rectified labels, which are refreshed each round. No additional cold-start training is required.
>
>
> We would be happy to address additional concerns and sincerely appreciate your thoughtful guidance on our paper.
>
> Sincerely,
>
> Paper 15295 Authors

---

### Official Review · Reviewer_oKDJ · 2025-10-31

**Soundness:** 3
**Presentation:** 3
**Contribution:** 2
**Rating:** 4
**Confidence:** 3

**Summary:**

This work is about learning from datasets that have long tailed class distributions and label noise. The method proposed is to leverage three complementary experts (text, image, and observed labels) with a class adaptive top K consensus mechanism to correct noisy labels. The central argument is that tail classes require more conservative consensus (smaller K) to avoid confirmation bias, while head classes can tolerate more greater consensus (larger K).

**Strengths:**

+ The identified problem is timely and interesting. The class agnostic label correction can actually harm performance by insufficiently correcting tail class labels.
+ The theoretical analysis is interesting and showing how consensus based refinement amplifies reliability. It provides good intuition for the design choices.
+ While missing some standard datasets in this task like CIFAR-10, the experiments are thorough enough, they spans on CIFAR-100-LTN, and mini-ImageNet-LTN, which are synthetic datasets, and real world datasets like Food101N, WebVision-50 under various noise types and imbalance factors. Results consistently show improvements, particularly notable gains for severe imbalance scenarios.
+ It is well presented.

**Weaknesses:**

- While the combination is novel and backed by the theoretical analysis, the individual components, like using CLIP for label correction, expert consensus, and top k voting have been explored previously . The main contribution is the class adaptive mechanism, which while effective, represents an incremental advance.
- Compared to the state of the art evaluations, real world noise evaluation is limited to webvision50 and food101n. Using Clothing1M would strengthen the claims.
- The method fundamentally relies on CLIP, limiting applicability to domains where such pre-trained models aren't available or perform poorly.
- The proof of Theorem 1 assumes conditional independence between pTE and pIE, but this assumption may not hold since both derive from CLIP's shared vision language space.

Minor:
- Eq 4 appears complex and could benefit from clearer notation.

**Questions:**

- how to handle edge cases e.g. when all experts disagree completely or when Kc becomes 0 for extremely rare classes?
- can you discuss failure cases or provide error analysis to understand when the proposed method might underperform?

---

> ### Author Response · Authors · 2025-11-20
>
> >**W1**: The main contribution is the class adaptive mechanism, which while effective, represents an incremental advance.
> ---
>
> **R1**: We thank the reviewer for the constructive feedback and accurate summary. We agree that our framework, CARE, thoughtfully integrates several established components, such as multi-source expert consensus and Top-K voting, which are grounded in prior work. However, the central motivation of our paper is that a naive, class-agnostic combination of these components is ineffective, or even detrimental, for the challenging Long-Tailed Noisy Label (LTNL) problem. As we demonstrate, class-agnostic strategies fail to address the uneven noise distribution, which disproportionately affects tail classes. Therefore, we respectfully argue that the class-adaptive mechanism is not an incremental advance but rather the essential and necessary component that enables this combination to succeed. This is empirically validated in our preliminary analysis in Figure 1 and Table 1. We show that class-agnostic label correction (RLD 1, RLD 2) provides minimal gain (79.6% vs 79.5%) or even degrades performance (76.8% vs 79.5%) compared to the original noisy labels. In contrast, our class-aware approach (RLD 3) achieves a clear performance improvement (80.9%). Our class-adaptive Top-K mechanism (which assigns a smaller K to tail classes)  is specifically designed to solve this non-trivial challenge. It is critical for extracting reliable labels in tail classes without over-filtering, a key failure point of class-agnostic methods. Thus, our main contribution is the novel class-aware mechanism that is critical to unlock the synergy between these experts, turning a failing strategy into an effective, state-of-the-art solution for the LTNL problem.
>
> >**W2**: Compared to the state of the art evaluations, real world noise evaluation is limited to webvision50 and food101n. Using Clothing1M would strengthen the claims.
>
> ---
>
> **R2**: Thank you for the suggestion. The experimental results of Clothing1M are presented as in Table R2.
> CARE yields a modest improvement on Clothing1M (71.2 vs. 71.4), which can be attributed to the dataset’s abundant training samples and atypical class imbalance.
> Specifically, even the least frequent classes contain 19,743 samples, which is substantially higher than the tail classes in typical long-tailed datasets, indicating that these classes are already well represented.
> Consequently, the impact of methods designed to address tail-data scarcity is limited.
> Furthermore, in Clothing1M, some classes that are rare in the test set actually have a relatively large number of training samples, which is fundamentally different from real-world long-tailed distributions.
> In typical long-tailed scenarios [R1, R5, R6], each class is considered equally important, and tail classes are consistently underrepresented both in training and testing, leading to a relatively balanced test set that reflects the importance of all classes.
> This contrasts with Clothing1M, where the training set abundance for some rare test classes reduces the effect of tail scarcity.
> Despite these factors, CARE still achieves a measurable gain, showing that the method remains effective even when the dataset deviates from the specific long-tailed, noisy conditions it is primarily designed for.
>
> **Table R2** Experimental results (%) of Clothing1M.
>
> | Datasets/Methods| without CARE | with CARE |
> | - | - | - |
> | Clothing1M      | 71.2         | 71.4      |
>
> **Reference**:
>
> [R1] Menon, et al. Long-tail learning via logit adjustment. _ICLR_, 2021.
>
> [R5] Boyan Zhou, Quan Cui, Xiu-Shen Wei, and Zhao-Min Chen. BBN: Bilateral-branch network with cumulative learning for long-tailed visual recognition. In _CVPR_, pp. 9719–9728, 2020.
>
> [R6] Mengke Li, HU Zhikai, Yang Lu, Weichao Lan, Yiu-ming Cheung, and Hui Huang. Feature fusion from head to tail for long-tailed visual recognition. In _AAAI_, volume 38, pp. 13581–13589, 2024b.

---

> ### Author Response · Authors · 2025-11-20
>
> >**W3**: The method fundamentally relies on CLIP, limiting applicability to domains where such pre-trained models aren't available or perform poorly.
>
> ---
>
> **R3**: We thank the reviewer for this important question regarding the method's generality. While we utilized CLIP in our main experiments to achieve state-of-the-art performance, the core mechanism of CARE is not fundamentally limited to CLIP. The framework is designed to leverage consensus and discrepancy from multiple expert sources (Text, Image, and Observed Label). To verify this, we conducted supplementary experiments in **Appendix E**, specifically addressing the reliance on powerful backbones. In this analysis, we replaced the CLIP-based experts with a standard ResNet backbone and GloVe text embeddings. The results, shown in Table 9, demonstrate that CARE still consistently surpasses the baseline model even with these less powerful components. For example, it improves accuracy from 58.9% to 61.1% on CIFAR100_IF10_NR50. As we discuss in **Appendix E**, this shows that CARE's effectiveness is not solely dependent on the absolute accuracy of the experts, but rather on its ability to leverage the diversity and disagreement in their predictions to perform robust label correction. Therefore, the CARE framework remains applicable so long as multiple, complementary expert signals can be obtained, even if they are not from a large-scale VLM.
>
> >**W4**: The proof of Theorem 1 assumes conditional independence between pTE and pIE, but this assumption may not hold since both derive from CLIP's shared vision language space.
>
> ---
>
> **R4**: To clarify, during training, the text expert model $p^{TE}$ is frozen, while the image expert model $p^{IE}$ is fine-tuned. Since $p^{TE}$ remains fixed and does not change during training, any updates to $p^{IE}$ do not influence  $p^{TE}$. Therefore, we believe it is reasonable to assume conditional independence between the two models in this context.
>
>
> >**W5**: Eq 4 appears complex and could benefit from clearer notation.
>
> ---
>
> **R5**: We appreciate the reviewer’s suggestion. To make Eq. 4 clearer, we have rewritten the update process using a unified expert-weighted formulation: $F_c^{(e)} =  F_c^{(e-1)} + \sum_{m \in \{\mathrm{TE},\,\mathrm{IE},\,\mathrm{BE}\}} \alpha_m(x)\, g_m(c),$, where the accumulated class frequency for each sample is expressed as a weighted sum of contributions from all experts. Each expert’s contribution is proportional to its confidence in the top-$K$ predicted classes and adjusted based on the reliability of the observed label. This simplification removes case-specific branches while remaining fully consistent with the implementation, making the update mechanism easier to understand. For further details, please refer to Pages 4-5 of the revised paper.

---

> ### Author Response · Authors · 2025-11-20
>
> >**Q1**: how to handle edge cases e.g. when all experts disagree completely or when Kc becomes 0 for extremely rare classes?
>
> ---
>
> **A1**: Thank you for these important questions about the method's robustness to edge cases. Our framework has specific mechanisms to handle both scenarios:
>
> 1. Handling Complete Disagreement (No Consensus):
>
>    This scenario, where the observed label (BE) is outside the Top-K predictions of the Text Expert (TE) and Image Expert (IE), is treated as a high-confidence signal of a noisy label. As described in our paper (e.g., in Equation 4 and the "no-consensus case" illustrated in Figure 2), in this situation, we do not reinforce or accumulate frequency for the observed label. Instead, the system is designed to "only accumulate the confidence scores corresponding to the top predicted categories of the experts [TE and IE]." This prevents the likely incorrect observed label from influencing the final decision and allows the system to gather evidence for a more plausible label based purely on the agreement (or lack thereof) between the cross-modal experts (TE and IE).
>
> 2. Handling $K_c=0$ for Extremely Rare Classes:
>
>    We proactively prevent $K_c$ from being zero by handling the class frequency ($n_c^e$) for classes with zero observed samples. In our implementation, if a class is so rare that its observed sample count, $n_c^e$, is zero in the initial data distribution, we do not use zero for the calculation of $K_c$. Instead, the effective sample count for that class is set to the minimum non-zero sample count found across all other classes in the dataset. This ensures that the effective $n_c^e$ for every class is at least 1. Since $K_c$ is calculated using a function of $n_c^e$ that employs the ceiling function (e.g., the Exponential form $\lceil(n_c^e)^{1/4}\rceil$ used in Algorithm 1 and discussed in **Appendix I**), this guarantees that the resulting $K_c$ is always $\ge 1$, thus allowing the class-adaptive Top-K consensus mechanism to operate for all defined classes.
>
> >**Q2**: can you discuss failure cases or provide error analysis to understand when the proposed method might underperform?
>
> ---
>
> **A2**: Since our method is primarily designed to address the challenges of tail classes, specifically their limited sample size and higher proportion of noisy labels, the multi-expert consensus mechanism helps reduce noise within tail classes and effectively increases the number of clean samples. However, when the dataset is relatively balanced, with no pronounced tail classes and only a low noise ratio in those classes, CARE can still slightly reduce the noise level, but this does not necessarily translate into improved tail-class performance or overall accuracy. For example, as shown in Table 2 under the imbalance factor of 10 and noise rate of 0.1, the performance of “CLIP+LA” and “CLIP+LA w/ CARE (ours)” shows a slight drop (84.0 vs. 83.9).

---

> ### Author Response · Authors · 2025-11-28
>
> Dear Reviewer oKDJ,
>
> We sincerely appreciate your constructive comments, which have significantly improved the quality and clarity of our work. Understanding the heavy workload of reviewers, and in order to facilitate a more efficient review, we have prepared a concise summary of our response below:
>
> - **Novelty**: While CARE uses known components (e.g., expert consensus), our class-adaptive Top‑K mechanism is essential, not incremental, for long-tailed noisy settings. Class-agnostic variants (RLD 1/2) fail (Table 1), while our adaptive design succeeds by targeting higher noise in tail classes (Fig. 1, Table R5).
> - **Results on Clothing1M**: We have added results (Table R2: 71.2 w.o. CARE → 71.4 w. CARE). The modest improvement is expected, as Clothing1M does not follow a typical long-tailed structure and many tail classes are still sufficiently sampled in the training set. Nevertheless, CARE still provides a consistent gain.
> - **CLIP dependence**: To verify this, we conducted supplementary experiments in **Appendix E**, specifically addressing the reliance on powerful backbones. In this analysis, we replaced the CLIP-based experts with a standard ResNet backbone and GloVe text embeddings.
> - **Independence assumption**: TE is frozen during training while IE is updated, making conditional independence reasonable in practice.
> - **No consensus**: When full disagreement among experts, we do not reinforce or accumulate frequency for the observed label. Instead, the system is designed to "only accumulate the confidence scores corresponding to the top predicted categories of the experts [TE and IE]."
>
> We would be glad to address any concerns and look forward to continued discussion. Thank you again for your valuable feedback on our paper.
>
> Sincerely,
>
> Paper 15295 Authors

---

### Official Review · Reviewer_JGXp · 2025-11-02

**Soundness:** 3
**Presentation:** 2
**Contribution:** 2
**Rating:** 6
**Confidence:** 4

**Summary:**

The paper presents "Class-Adaptive Rectification with Experts (CARE)," a framework for robust learning in the context of long-tailed noisy label (LTNL) problems. It addresses the challenges of label noise and class imbalance, particularly in tail classes, by leveraging a class-aware Top-K expert consensus mechanism. The proposed method integrates noisy labels, text embeddings, and image features to improve label correction and class frequency estimation. Experimental results on multiple benchmarks, including CIFAR-100-LTN and real-world datasets, show that CARE consistently outperforms existing state-of-the-art methods, achieving significant accuracy improvements. The authors provide a comprehensive analysis of their approach and its implications for future research.

**Strengths:**

Innovative Approach: The CARE framework effectively combines label rectification with class-adaptive techniques, addressing both noisy labels and long-tailed distributions.

Comprehensive Evaluation: The framework is tested on various benchmarks, demonstrating consistent performance improvements over state-of-the-art methods.

Strong Theoretical Basis: The paper provides solid theoretical justifications for its design choices, enhancing its credibility and understanding.

**Weaknesses:**

1. The CARE framework may require significant computational resources due to the integration of multiple expert models, which could limit its practicality in resource-constrained environments.

2. The evaluation primarily focuses on specific datasets, raising concerns about the framework's adaptability to other domains or types of noise not covered in the experiments.

3. The emphasis on correcting tail class labels could lead to overfitting, especially if the tail classes are too small or underrepresented in the training data.

4.  The reliance on accuracy as the sole performance metric may overlook other critical aspects of model performance, such as precision, recall, or F1-score, particularly in imbalanced datasets.

**Questions:**

How does CARE perform with larger datasets?

Can it handle other noise types or distributions?

What can be done to prevent overfitting on tail classes?

---

> ### Author Response · Authors · 2025-11-20
>
> > **W1**: The CARE framework may require significant computational resources due to the integration of multiple expert models, which could limit its practicality in resource-constrained environments.
>
> ---
>
> **R1**: We thank the reviewer for raising this practical concern regarding computational resources. We designed CARE specifically as a parameter-efficient framework to minimize this issue. The computational overhead from the integrated experts is minimal. This is because the Text Expert (TE) is a pre-trained VLM that remains frozen and requires no additional fine-tuning. Its only cost is a forward pass to generate semantic predictions. The Observed-Label Expert (BE) is simply the given annotation and adds no computational cost. Crucially, the Image Expert (IE) is not a separate, additional model that needs training. It is the main classification network itself, which is being fine-tuned for the task (using AdaptFormer). We simply reuse the outputs from this main model for the consensus phase. As the paper notes, this component "remains parameter-free during the expert consensus phase". Therefore, the only significant added cost is one forward pass from the frozen TE, not the training of multiple, large expert models.
>
> > **W2**: The reliance on accuracy as the sole performance metric may overlook other critical aspects of model performance, such as precision, recall, or F1-score, particularly in imbalanced datasets.
>
> ---
>
> **R2**: Macro F1-score is the average of the F1-score computed independently for each class, giving equal weight to all classes regardless of their size.
> By combining precision and recall, Macro F1-score captures how well a model balances false positives and false negatives across classes.
> This makes it sensitive to class imbalance, as poor performance on minority classes cannot be offset by strong performance on majority classes.
>
> **Table R1.** The Macro F1-score (%) across multiple datasets.
>
> |        Datasets/Methods              | without CARE | with CARE |
> | - | - | - |
> | CIFAR100_IF100_NR30(Joint Noise)     | 77.2         | 78.3      |
> | CIFAR100_IF100_NR50(Joint Noise)     | 75.0         | 76.4      |
> | CIFAR100_IF10_NR40(Symmetric Noise)  | 80.3         | 81.4      |
> | CIFAR100_IF10_NR60(Symmetric Noise)  | 75.7         | 78.9      |
> | CIFAR100_IF10_NR20(Asymmetric Noise) | 80.0         | 80.7      |
> | CIFAR100_IF10_NR40(Asymmetric Noise) | 67.6         | 69.7      |
>
> We evaluated our method in terms of Macro F1-score(\%) across multiple datasets, as shown in Table R1.
> CARE consistently improves Macro F1-score under all noise settings. For joint noise (NR30/NR50), it achieves gains of approximately 1-1.4 points.
> For symmetric noise (NR40/NR60), CARE achieves consistent improvements, including a substantial 3.2-point increase at $NR=60$.
> Under the more challenging asymmetric noise scenario (NR20/NR40), CARE continues to demonstrate measurable performance gains.
> Overall, CARE reliably enhances class-balanced performance across different noise types and severities.

---

> ### Author Response · Authors · 2025-11-20
>
> >**Q1**: How does CARE perform with larger datasets?
> ---
>
> **A1**: Thank you for the suggestion. We conducted additional experiments on the larger Clothing1M dataset, and the experimental results are presented in Table R2.
> CARE yields a modest improvement on Clothing1M (71.2 vs. 71.4), which can be attributed to the dataset’s abundant training samples and atypical class imbalance.
> Specifically, even the least frequent classes contain 19,743 samples, which is substantially higher than the tail classes in typical long-tailed datasets, indicating that these classes are already well represented.
> Consequently, the impact of methods designed to address tail-data scarcity is limited.
> Furthermore, in Clothing1M, some classes that are rare in the test set actually have a relatively large number of training samples, which is fundamentally different from real-world long-tailed distributions.
> In typical long-tailed scenarios [R1, R5, R6], each class is considered equally important, and tail classes are consistently underrepresented both in training and testing, leading to a relatively balanced test set that reflects the importance of all classes.
> This contrasts with Clothing1M, where the training set abundance for some rare test classes reduces the effect of tail scarcity.
> Despite these factors, CARE still achieves a measurable gain, showing that the method remains effective even when the dataset deviates from the specific long-tailed, noisy conditions it is primarily designed for.
>
> **Table R2.** Experimental results (%) of Clothing1M.
>
> | Datasets/Methods| without CARE | with CARE |
> | - | - | - |
> | Clothing1M      | 71.2         | 71.4      |
>
>
> **Reference**:
>
> [R1] Menon, et al. Long-tail learning via logit adjustment. ICLR, 2021.
>
> [R5] Boyan Zhou, Quan Cui, Xiu-Shen Wei, and Zhao-Min Chen. BBN: Bilateral-branch network with cumulative learning for long-tailed visual recognition. In CVPR, pp. 9719-9728, 2020.
>
> [R6] Mengke Li, HU Zhikai, Yang Lu, Weichao Lan, Yiu-ming Cheung, and Hui Huang. Feature fusion from head to tail for long-tailed visual recognition. In AAAI, volume 38, pp. 13581-13589, 2024b.
>
> > **Q2**: Can it handle other noise types or distributions?
> ---
>
> **A2**: We evaluated two additional noise types across multiple datasets:
>
> 1. **Standard Symmetric Noise (Uniform Noise):** Its transition matrix is defined as
>    T_ii=1−η (probability of keeping the correct label),
>    T_ij=η/(num_classes−1)for i≠j (probability of flipping from class i to class j).
> 2. **Asymmetric Pair-Flip Noise:** Instead of flipping to any random class, each class is flipped to one specific, visually similar class, modeling realistic inter-class confusion.
>
> The experimental results are summarized in Table R3.
>
> **Table R3.** Experimental results (%) of two additional noise types.
>
> |        Datasets/Methods               | without CARE | with CARE |
> | - | - | - |
> | CIFAR100_IF100_NR40 (Uniform Noise)   | 70.8         | 74.1      |
> | CIFAR100_IF100_NR60 (Uniform Noise)   | 65.3         | 69.3      |
> | CIFAR100_IF100_NR20 (Pair-Flip Noise) | 77.9         | 79.4      |
> | CIFAR100_IF100_NR40 (Pair-Flip Noise) | 67.0         | 70.7      |
>
>
> ------
>
> Table R3 demonstrates that CARE consistently improves performance under both standard symmetric noise and asymmetric pair-flip noise. For standard symmetric noise (NR40 and NR60), CARE yields gains of 3.3 and 4.0 points, respectively, indicating that it effectively mitigates randomly distributed label corruption. For pair-flip noise (NR20 and NR40), which is more structured and challenging due to class similarity, CARE still delivers improvements of 1.5–3.7 points. Overall, the results show that CARE generalizes well across different noise types, enhancing robustness in both random and structured label noise settings.
>
> >**Q3**: What can be done to prevent overfitting on tail classes?
> ---
>
> **A3**: Thank you for the insightful comment. While tail classes indeed contain fewer samples and could in principle be more prone to overfitting, our method mitigates this risk in several ways. First, CARE does not rely on a single model’s prediction for tail classes; instead, it aggregates multi-expert consensus together with CLIP-based semantic validation, which serves as a strong external regularizer and prevents the model from memorizing noisy tail labels. Second, the class-adaptive mechanism effectively increases the number of reliable tail-class samples through label correction, which helps reduce the risk of overfitting to only a few rare instances.
>
> To further mitigate potential overfitting, one could (and we plan to explore in future work):
>
> - Applying stronger data augmentation or Mixup specifically to tail classes,
> - Using class-balanced loss or re-weighting to avoid over-emphasizing a small set of corrected samples,
> - Introducing stricter confidence thresholds so that tail-class labels are updated only when multi-expert consensus is highly reliable.

---

> > ### Comment · Reviewer_JGXp · 2025-11-24
> >
> > Thanks for the rebuttal. I am going to keep the initial score and wait for the feedback from other reviewers.

---

> ### Author Response · Authors · 2025-11-25
>
> Dear Reviewer JGXp
>
> Thanks for your recognition of our responses and for maintaining a positive rating.
>
> If there are any remaining concerns or points that would benefit from further clarification, we would be grateful for the opportunity to provide additional details or discuss them further.
>
> Sincerely,
>
> Paper 15295 Authors

---

### Author Response · Authors · 2025-11-30
**Rebuttal Summary**

Dear AC,

We thank all reviewers for their valuable and constructive feedback. Our paper proposes CARE, a novel framework that addresses the challenging problem of Learning with Long-Tailed Noisy Labels (LTNL). At its core is a Class-Adaptive Rectification mechanism that dynamically adjusts label correction intensity according to class frequency.

This work identifies a key weakness in existing label-correction methods for LTNL: their class-agnostic design is ineffective under realistic, uneven noise distributions. Our **primary contribution** is a new class-adaptive Top-$K$ consensus mechanism, which strategically combines predictions from the image expert, the text expert, and the observed labels. Integrated throughout the CARE framework, this mechanism customizes correction strictness based on class frequency, directly targeting the high noise rates prevalent in tail classes. It serves as the essential component that turns an otherwise ineffective, class-agnostic approach into a state-of-the-art solution, as clearly supported by our preliminary analysis in Figure 1 and Table 1.

The key strengths emphasized in our rebuttal are summarized below:

**Effectiveness**: CARE delivers consistent state-of-the-art performance across multiple benchmarks (CIFAR, Mini-ImageNet, WebVision, Food101-N) and under various noise settings (symmetric, asymmetric, pair-flip). It significantly enhances accuracy on critical tail classes while maintaining strong head-class performance.

**Theoretical and Empirical Rigor**: We provide a formal proof for Proposition 3 in Appendix D, along with comprehensive ablation studies (e.g., Table R4). These validate our central claim that assigning a smaller K to tail classes is vital for achieving higher consensus precision.
Generality: Although optimal performance is achieved with CLIP, CARE is not fundamentally dependent on it. Experiments in Appendix E show that CARE still brings substantial improvements even when using weaker backbones such as ResNet and text embeddings like GloVe, demonstrating its general applicability.

**Practicality**: The framework is designed to be parameter-efficient. The Text Expert remains frozen, and the Image Expert is the primary fine-tuned model, minimizing computational overhead. Its usability with standard backbones such as ResNet has been empirically verified.

In summary, CARE establishes a novel and essential class-adaptive paradigm for LTNL, supported by solid theoretical foundations and extensive experimental validation. It represents an effective and practical solution for this challenging real-world problem.

Sincerely,

Paper 15295 Authors

---

### Meta-Review · Area_Chair_MYfn · 2025-12-31

**Summary:**

The paper proposes CARE, a framework for Learning with Long-Tailed Noisy Labels (LTNL) that utilizes a class-adaptive Top-$K$ expert consensus mechanism involving text, image, and observed label experts. The primary concerns raised by reviewers included: (1) a lack of empirical and theoretical justification for why tail classes require more stringent (smaller $K$) correction; (2) potential "incremental" novelty, as components like CLIP and expert consensus are known; (3) reliance on specific VLM backbones (CLIP); and (4) technical clarity regarding the training pipeline, parameter initialization, and the mathematical proof of Proposition 3.

**Reviewer Concerns:**

*   **Addressed by Rebuttal:**
    *   The authors successfully addressed Reviewer oKDJ’s concern by providing experiments in Appendix E using ResNet and GloVe, demonstrating that CARE still outperforms baselines without high-end VLM backbones.
    *   Authors provided Macro F1-scores (Table R1) and added results for the Clothing1M dataset (Table R2), satisfying the requests of Reviewers JGXp and oKDJ.
    *  The authors clarified that the framework is parameter-efficient, reusing the main classification network as the Image Expert and keeping the Text Expert frozen, which addressed Reviewer JGXp’s concerns.
    *   The authors provided a revised formal proof for Proposition 3 and clarified the initialization of the AdaptFormer and classifier weights (initialized via text features), which addressed the technical confusion from Reviewer oswe.
    *  The addition of Table R4 (comparing global $K$ vs. class-wise $K_c$) provided the necessary ablation requested by Reviewer oswe to prove the specific benefit of the adaptive strategy.

*   **Outstanding:**
    *    While the authors argued that the class-adaptive mechanism is the "essential component" that makes the combination of experts work, Reviewer oKDJ may still view the integration of existing components as an incremental advance.
    *    Reviewer oKDJ pointed out potential conditional dependence between the Text and Image experts since both originate from CLIP space. While the authors argued that the freezing vs. fine-tuning creates independence in practice, this remains a point of theoretical debate.

**Reviewer Scores:**

**Reviewer oswe (Initial: 2):** **Predicted Final: 4.** The authors provided answers to this reviewer's questions, including the missing ablation on $K$ and a revised proof.

---

### Decision · Program_Chairs · 2026-01-26

Reject